# Local polar order controls mechanical stress and triggers layer formation in *Myxococcus xanthus* colonies

Endao Han [1,2] ✉, Chenyi Fei[3,4], Ricard Alert [5,6,7], Katherine Copenhagen[3], Matthias D. Koch [3,4], Ned S. Wingreen [3,4] & Joshua W. Shaevitz [1,3] ✉

Colonies of the social bacterium *Myxococcus xanthus* go through a morphological transition from a thin colony of cells to three-dimensional droplet-like fruiting bodies as a strategy to survive starvation. The biological pathways that control the decision to form a fruiting body have been studied extensively. However, the mechanical events that trigger the creation of multiple cell layers and give rise to droplet formation remain poorly understood. By measuring cell orientation, velocity, polarity, and force with cell-scale resolution, we reveal a stochastic local polar order in addition to the more obvious nematic order. Average cell velocity and active force at topological defects agree with predictions from active nematic theory, but their fluctuations are substantially larger than the mean due to polar active forces generated by the self-propelled rod-shaped cells. We find that *M. xanthus* cells adjust their reversal frequency to tune the magnitude of this local polar order, which in turn controls the mechanical stresses and triggers layer formation in the colonies.

Biological cells often form densely-packed, two-dimensional monolayers that serve specific biological functions. Densely-packed cells with elongated shapes, from collectives of eukaryotes[1–8] to populations of bacteria[9–14], typically align with each other and may behave as active nematic liquid crystals. The constituents of such active nematics generate internal active stresses along the axis of alignment, which give rise to phenomena not found in their passive counterparts[15,16]. A hallmark of these systems is the spontaneous creation of topological defects–singularities in the orientation field that play an important role in apoptotic cell extrusion[2], the accumulation of neural progenitor cells[8], tissue morphogenesis[17,18], and pattern formation in bacterial colonies[10–14].

However, motile cells are propelled by an active polar force, and hence polar order can contribute to their collective dynamics. For example, epithelial cell layers can develop polar order, which drives flocking, morphogenesis at defects, and spreading[18–20]. Motile bacteria are also driven by polar forces produced by flagella, pili, or gliding motors. What role does this single-cell polarity have in the collective dynamics? This question has been addressed using the Self-Propelled Rod (SPR) model[21,22], in which the interactions between rods are either apolar or only weakly polar.

Interestingly, both nematic order and polar order emerge in SPR systems. Some SPRs show long-range nematic order only[23,24] while others show a mixture of nematic and polar order[25,26]. It is not always obvious which kind of order plays a more important role in governing the dynamics of a given system and in controlling the corresponding biological functions. For example, the formation of cell aggregates and layered structures in colonies of self-propelled rod-like bacteria has been explained in different ways, such as by modeling the system as SPRs[27,28], active nematics[14], or both[13]. Thus, the coexistence and interplay between nematic and polar order remain poorly understood.

[1]Joseph Henry Laboratories of Physics, Princeton University, Princeton, NJ, USA. [2]School of Physical and Mathematical Sciences, Nanyang Technological University, Singapore, Singapore. [3]Lewis-Sigler Institute for Integrative Genomics, Princeton University, Princeton, NJ, USA. [4]Department of Molecular Biology, Princeton University, Princeton, NJ, USA. [5]Max Planck Institute for the Physics of Complex Systems, Dresden, Germany. [6]Center for Systems Biology Dresden, Dresden, Germany. [7]Cluster of Excellence Physics of Life, TU Dresden, Dresden, Germany. ✉e-mail: endao.han@ntu.edu.sg; shaevitz@princeton.edu

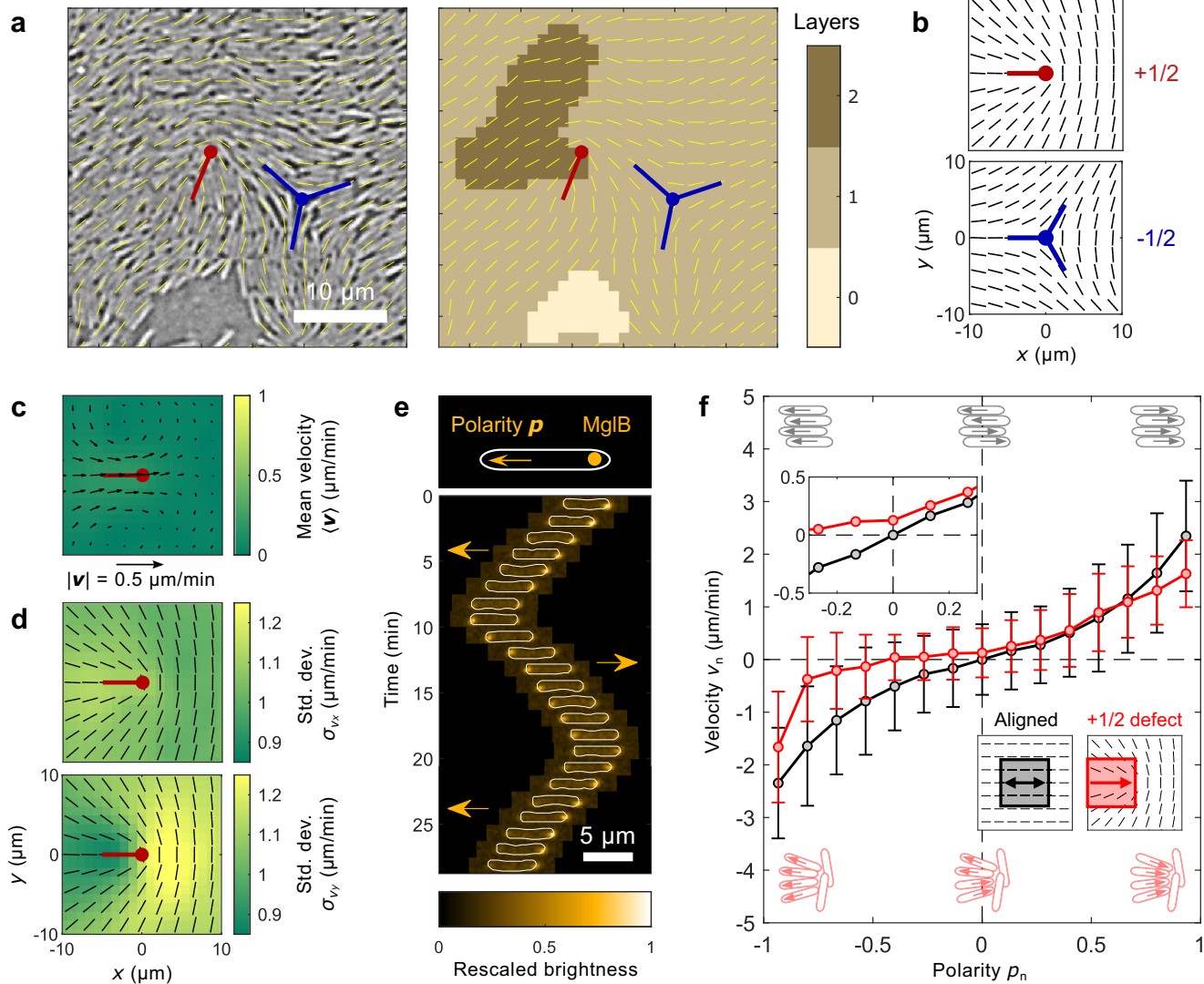

**Fig. 1 | Nematic and polar order in thin *M. xanthus* layers. a** An exemplary bright field image of cells on a solid surface and the corresponding number of cell layers (see SI Fig. S2). Yellow line segments indicate the director field (cell orientations) $\hat{n}$, and there is a pair of +1/2 (red) and −1/2 (blue) defects in the view. The white scale bar is 10 μm. **b** Average director field $\langle \hat{n} \rangle$ around ±1/2 defects. **c** Mean velocity of cell flow $\langle v \rangle$ around +1/2 defects; the black arrows show magnitude and direction and the color map shows the speed $|\langle v \rangle|$. **d** Standard deviation $\sigma_{v_x}$ and $\sigma_{v_y}$ of the $x$ and $y$ components of the velocity field around +1/2 defects. The black line segments show $\langle \hat{n} \rangle$. **e** MglB proteins localize to the rear of the cell and define individual cell polarity (arrows). The kymograph shows the brightness of the fluorescent tag for one isolated moving cell (outlined by the white contours). The kymograph captures two reversal events. The white scale bar is 5 μm. **f** Local cell velocity $v_n$ is correlated with local polarity $p_n$. Black circles indicate measurements in nematically aligned regions (in the black square) while red circles are for measurements in the comet tail region of +1/2 defects (in the red square). The data were measured with 7 samples and the error bars show the standard deviation. The inset is a zoom-in of the region near $p_n = 0$. The positive direction for polarity and velocity are labeled by the thick arrows in the director field: it points toward the defect core near +1/2 defects and is left-right symmetric in aligned regions.

In this work, we investigate the interplay of nematic and polar order in colonies of the soil bacterium *Myxococcus xanthus*. This rod-like bacterium glides on surfaces, and it stochastically reverses its direction of motion with an average reversal time $\tau$ that is approximately independent of the cell density in the colony[29,30]. In a nutrient-rich environment, $\tau \approx 10$ min on average[30,31] and the cells spread into a thin layer (Fig. 1a). As nutrients become scarce, the cells increase $\tau$ and start to form second or third layers on top of the original monolayer[30,32]. Layer formation initiates a developmental process leading to three-dimensional fruiting bodies inside of which cells sporulate[30,33,34]. Previous work showed that layer formation is promoted by topological defects in the cell alignment field, and explained the observations by modeling the colony as an active nematic liquid crystal[14]. This active nematic model applies at time scales significantly longer than $\tau$. However, at shorter times, cell reversals could produce polarity fluctuations, whose effects on layer formation remain unclear.

Here, we show that polarity fluctuations trigger the formation of new cell layers at topological defects, thus initiating the transition from monolayers to droplet-like fruiting bodies in *M. xanthus* colonies. The average flow and traction fields around topological defects are well captured by the active nematic model. However, fluctuations in local polarity produce traction fluctuations much stronger than the average, which trigger the formation of new cell layers. We also find that the decreasing cell reversal frequency produces stronger traction fluctuations, which promotes the formation of new layers. This finding suggests that *M. xanthus* may control their reversal rate to trigger changes in colony morphology.

## Results

### Instantaneous cell polarity is the main driver of cell flow

Layers of *M. xanthus* cells contain both comet-like defects of topological charge +1/2 and triangular defects of charge −1/2 (Fig. 1a, b), indicating that the system is nematic. Previous work showed that the average cell flow around these defects is well-explained by an active nematic model[14] (see SI Fig. S1). Figure 1c shows the mean cell velocity $\langle \boldsymbol{v} \rangle$ near a +1/2 defect, where the symbol "$\langle \rangle$" represents a temporal average in the comoving frame of all the defects tracked. It originates from the balance between the active nematic force density,

$$\boldsymbol{f}_n^a = -\zeta_n \nabla \cdot \boldsymbol{Q}, \tag{1}$$

and an anisotropic cell-substrate friction, where $\boldsymbol{Q} = S(2\hat{\boldsymbol{n}}\hat{\boldsymbol{n}} - \boldsymbol{I})$ is the nematic order parameter tensor, $\hat{\boldsymbol{n}}$ is the director of the nematic order, $S$ is the scalar order parameter, and $\zeta_n$ is the activity coefficient[14,15]. The active stress is extensile if $\zeta_n > 0$ and contractile if $\zeta_n < 0$. The velocity flux through a circular boundary $\mathcal{C}$ around a +1/2 defect, $\Phi_{\boldsymbol{v}} = \oint_{\mathcal{C}} (\boldsymbol{v} \cdot \hat{\boldsymbol{r}}) ds$, is negative, indicating a net influx of cells toward the defect core. Consequently, +1/2 defects promote cell accumulation and layer formation (Fig. 1a). Similarly, −1/2 defects produce a net outflux of cells and favor the formation of holes within the monolayer[14]. Thus, the nematic order is connected to colony morphology.

While the active nematic theory predicts a steady flow around +1/2 defects, the nucleation of new layers is rare, sudden, and stochastic. To study these stochastic events, we measured the velocity fluctuations. Strikingly, the standard deviations of $v_x$ and $v_y$, $\sigma_{v_x}$ and $\sigma_{v_y}$, are both several times larger than the mean speed $|\langle \boldsymbol{v} \rangle|$ (Fig. 1d). We hypothesized that these velocity fluctuations are driven by fluctuations in local polar order around +1/2 defects. To test this hypothesis, we experimentally measured cell polarity and traction forces with cellular-scale resolution simultaneously with the cell velocity, nematic order, and thickness fields near defects.

We measured cell polarity, $\boldsymbol{p}$, using the *mglB::mVenus* strain of *M. xanthus*. This strain expresses a fluorescent fusion to the MglB protein, which is localized to the rear pole of the cell (Fig. 1e)[35]. By simultaneously imaging the cells and the localization of MglB, we defined the polarity of each cell within the population (see SI Section V). We calculated the average local polarity $p_n$ and velocity $v_n$ in square regions with a side $l_p = 12\,\mu\text{m}$, the length of two cells, where the subscript $n$ denotes the component projected along the director. To obtain $p_n$ and $v_n$, we reoriented the regions either with aligned cells or near a +1/2 defect as shown in Fig. 1f, and averaged the horizontal components of $\boldsymbol{p}$ and $\boldsymbol{v}$, respectively inside the boxes (see SI Fig. S2). For the +1/2 defect, "right" was defined as the positive direction, while due to the left-right symmetry of the aligned region, $(p_n, v_n)$ and $(-p_n, -v_n)$ are equivalent. In the areas with aligned cells ($\nabla \cdot \boldsymbol{Q} \to 0$), $\boldsymbol{f}_n^a$ does not drive any consistent cell flow, and the cell flow is driven by instantaneous polarity. In these regions, $v_n$ is a monotonic function of $p_n$ (Fig. 1f, black), and the slope of $v_n$ versus $p_n$ increases as $|p_n| \to 1$, indicating that the increase of local polar order leads to higher cell speed. In contrast, near a +1/2 defect, both the nematic and polar order drive cell motion. Yet, instantaneous local polar order is still the major driver of the velocity field as seen in the monotonic $v_n$–$p_n$ relationship (Fig. 1f, red). However, compared to the aligned areas, this specific arrangement of cells around defects limits the local cell speed, leading to lower $v_n$ as $|p_n| \to 1$. Furthermore, at zero polarity $p_n = 0$, the cell velocity is non-zero (Fig. 1f inset, red); this is the flow driven by the net active nematic force $\boldsymbol{f}_n^a$ (Fig. 1c). Our results show that this average velocity due to the nematic active force $\boldsymbol{f}_n^a$ is small compared to the velocities produced by polarity fluctuations (Fig. 1f). These strong polarity fluctuations could be due to the sudden polarity reversals of *M. xanthus* cells (Fig. 1e).

### Polarity-driven cell influx drives layer formation

Having shown that polarity fluctuations are the main driver of instantaneous cell flows, we now show that these flows provide the main contribution to layer formation. The cell number was approximately constant as the cells did not grow or divide under our experimental conditions (see Methods). The formation of a second layer on top of a monolayer thus requires a local influx of cells via motility. The change in volume $\Delta V$ within a region is given by the velocity flux,

$$\Delta V = -\int \Phi_{\boldsymbol{v}} \, h \, dt, \tag{2}$$

where $h \approx 0.5\,\mu\text{m}$ is the thickness of a single cell layer (see SI Figs. S3 and S4). Figure 2a illustrates two possible scenarios leading to a net influx of cells: (i) an average flow (Fig. 1c) driven by the nematic active force (Eq. (1)) and (ii) stochastic cell movement produced by polarity fluctuations.

Figure 2b shows two circular regions with radius $l_p$, each surrounding a +1/2 defect. A visible second layer appeared at $t = 0$ min in region A, while region B remained as a monolayer. Figure 2c shows the polarity flux across the boundaries of these two regions $\Phi_{\boldsymbol{p}} = \oint_{\mathcal{C}} (\boldsymbol{p} \cdot \hat{\boldsymbol{r}}) ds$. For A, $\Phi_{\boldsymbol{p}}$ was consistently negative starting from several minutes before the out-of-plane cell motion, while $\Phi_{\boldsymbol{p}}$ for region B fluctuated around 0. The resulting velocity flux $\Phi_{\boldsymbol{v}}$ showed the same trends, with a net influx ($\Phi_{\boldsymbol{v}} < 0$) for the region A but not for B. We identified multiple regions around +1/2 defects with and without layer formation, and show the mean polarity flux $\overline{\Phi_{\boldsymbol{p}}}$, velocity flux $\overline{\Phi_{\boldsymbol{v}}}$, and volume change $\overline{\Delta V}$ in Fig. 2d, where the overline represents the sample mean. Their trends are consistent with the exemplary individual events in Fig. 2c, where $\Phi_{\boldsymbol{p}}$ and $\Phi_{\boldsymbol{v}}$ are directly related to layer formation.

Besides the average instantaneous flux $\overline{\Phi_{\boldsymbol{v}}}$, we calculated the flux based on the mean velocity $\langle \boldsymbol{v} \rangle$ near +1/2 defects (Fig. 1c), and obtained $\Phi_{\langle \boldsymbol{v} \rangle} = -5.1\,\mu\text{m}^2/\text{min}$. In $\Phi_{\langle \boldsymbol{v} \rangle}$, $\langle \boldsymbol{v} \rangle$ was averaged over all the +1/2 defects, with or without second layer formation, while in reporting $\overline{\Phi_{\boldsymbol{v}}}$, we selected two separate subsets: those with second layer formation and those without. Note that $\Phi_{\langle \boldsymbol{v} \rangle}$ is significantly weaker than the value of $\overline{\Phi_{\boldsymbol{v}}}$ leading to layer formation (Fig. 1d, purple). Similarly, if the cell accumulation was driven by the average flow $\langle \boldsymbol{v} \rangle$, the volume change $\Delta V(t)$ should follow the black dot-dashed line in Fig. 2d bottom. The measured volume change (purple in Fig. 1d) clearly departs from the predictions based on the average flow. Therefore, most of the influx at +1/2 defects is due to the strong velocity fluctuations induced by local polarity. We have therefore revealed the fluctuations that drive the onset of layer formation, which is a fast discontinuous process.

Figure 2e shows the distributions of $\Phi_{\boldsymbol{p}}$ and $\Phi_{\boldsymbol{v}}$ in cases with (purple) and without (green) layer formation. The two peaks are more clearly separated for the polarity flux $\Phi_{\boldsymbol{p}}$ than for the velocity flux $\Phi_{\boldsymbol{v}}$. These two fluxes do not always have a strong positive correlation (Fig. 2c, d). In cases of layer formation (purple in Fig. 2d), the velocity flux often touches zero whereas the polarity flux is well below zero consistently. This slight difference is because the velocity of a cell in the colony is not only determined by its polarity but also by cell-cell interactions.

### Cellular tractions and layer formation

To probe cellular forces in *M. xanthus* colonies, we used traction force microscopy (TFM)[36–38] to measure the traction $\boldsymbol{T}$, which is the shear stress exerted on the substrate by the cells (Fig. 3a). By tracking the fluorescent particles embedded in the substrate, we measured the displacements of the substrate tangential to its surface and we obtained the traction $\boldsymbol{T}$ (see SI Section VI). Since the cells glide on both the substrate and on neighboring cells, the total active force density has a cell-substrate contribution $\boldsymbol{f}_s^a$ and a cell-cell contribution $\boldsymbol{f}_c^a$. Similarly, the friction has a cell-substrate contribution $\boldsymbol{f}_s^f$ and a cell-cell contribution $\boldsymbol{f}_c^f$. The traction is the total force applied on the substrate

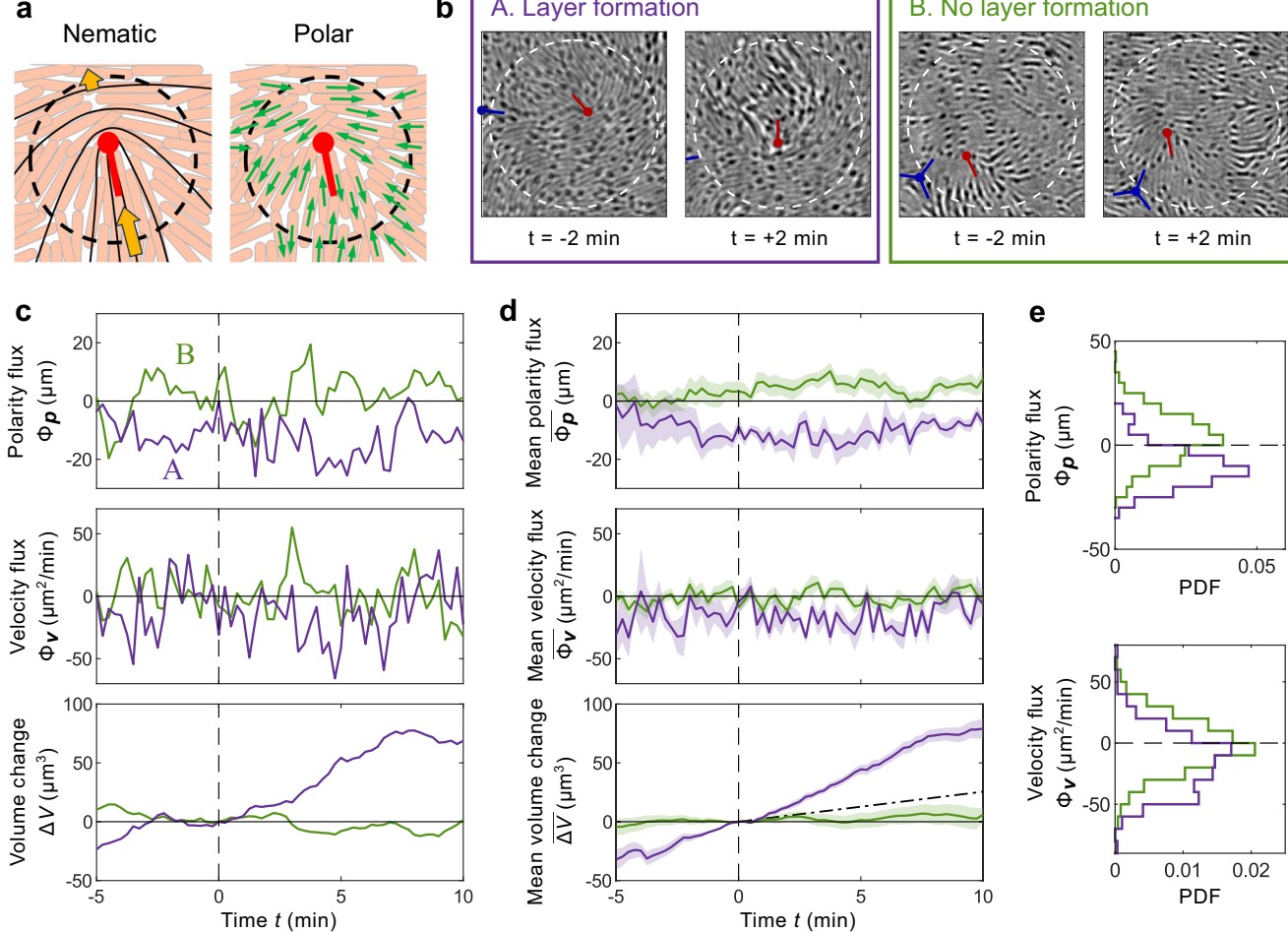

**Fig. 2 | Instantaneous cell flux drives layer formation. a** Schematic illustrations of cell flows driven by the nematic order (left) and polarity (right) around the same +1/2 defect. The dashed circle shows the boundary at which we calculate the velocity and polarity fluxes. The orange arrows show the cell velocity driven by the nematic active force, which is asymmetric due to anisotropic friction[14]. The solid black curves illustrate the director field near this defect. The green arrows label the cell polarity. **b** Two exemplary regions with (A, purple) and without (B, green) second layer formation. For region A, we set the time at which the second layer appeared as $t = 0$ min, while for B, the time point $t = 0$ min was chosen arbitrarily. The white circles with a radius of $l_p = 12$ μm label the boundaries of the selected regions, and they each surround a +1/2 defect. **c** Polarity flux $\Phi_{\boldsymbol{p}}$, velocity flux $\Phi_{\boldsymbol{v}}$, and volume change $\Delta V$ in regions A (purple) and B (green). **d** Mean (curves) polarity flux $\overline{\Phi_{\boldsymbol{p}}}$, velocity flux $\overline{\Phi_{\boldsymbol{v}}}$, and volume change $\overline{\Delta V}$ over multiple regions, each surrounding a +1/2 defect, with (purple) and without (green) second layer formation, and the corresponding standard errors (shaded areas). The expected volume change driven by the mean velocity (Fig. 1c) is shown by the black dot-dashed line, calculated using the experimentally measured $\Phi_{\langle \boldsymbol{v} \rangle}$ and Eq. (2). The definition of $t = 0$ min remains the same. **e** Probability density functions of the polarity flux $\Phi_{\boldsymbol{p}}$ and velocity flux $\Phi_{\boldsymbol{v}}$ within the time window of [−5, 10] min.

by the cells, thus

$$\boldsymbol{T} = - \left( \boldsymbol{f}_s^a + \boldsymbol{f}_s^f \right). \tag{3}$$

In the low-Reynolds-number regime, force balance is given by

$$\boldsymbol{f}_s^a + \boldsymbol{f}_c^a + \boldsymbol{f}_s^f + \boldsymbol{f}_c^f - \nabla P = 0, \tag{4}$$

and hence the traction can also be interpreted as the force density due to cell-cell interactions

$$\boldsymbol{T} = \boldsymbol{f}_c^a + \boldsymbol{f}_c^f - \nabla P, \tag{5}$$

where $P$ is the in-plane pressure within the cell layer (see SI Section II). We first focused on the average traction field $\langle \boldsymbol{T} \rangle$ around ±1/2 defects (Fig. 3b), which is $\langle \boldsymbol{T} \rangle = \langle \boldsymbol{f}_c^a \rangle + \langle \boldsymbol{f}_c^f \rangle - \nabla \langle P \rangle$ according to Eq. (5). To interpret this result, we made two assumptions: (1) when averaged over long time scales, cell layers behave like active nematics, so the cell-cell active force $\langle \boldsymbol{f}_c^a \rangle$ is given by $\boldsymbol{f}_n^a$ (Eq. (1)); and (2) the mean cell-

cell friction $\langle \boldsymbol{f}_c^f \rangle$ is negligible compared to cell-substrate friction[14] (see SI Section II). With these assumptions, we predict the mean traction field to be

$$\langle \boldsymbol{T} \rangle = - \zeta_n \nabla \cdot \langle \boldsymbol{Q} \rangle - \nabla \langle P \rangle, \tag{6}$$

which agrees with the experimental measurements (Fig. 3b). See SI Fig. S1 for detailed comparisons between experimental and theoretical traction and velocity fields, and SI Section II for the detailed fitting process. We obtained a positive activity, $\zeta_n > 0$, which agrees with the previous work showing that the cell layer is an extensile active nematic fluid[14]. As predicted from Eq. (6), the average traction $\langle \boldsymbol{T} \rangle$ is stronger near the cores of the defects, where the director field exhibits strong distortions (Fig. 1b).

However, beyond the average, our experiments show that, similar to the velocity field $\boldsymbol{v}$ (Fig. 1d), the instantaneous traction $\boldsymbol{T}(t)$ had a standard deviation $\sigma_T$ about an order of magnitude larger than the mean $|\langle \boldsymbol{T} \rangle|$ (Fig. 3c). Unlike $\langle \boldsymbol{T} \rangle$, the fluctuations were not controlled by the nematic order $\boldsymbol{Q}$. In regions where the cells were aligned ($\nabla \cdot \boldsymbol{Q} \to 0$),

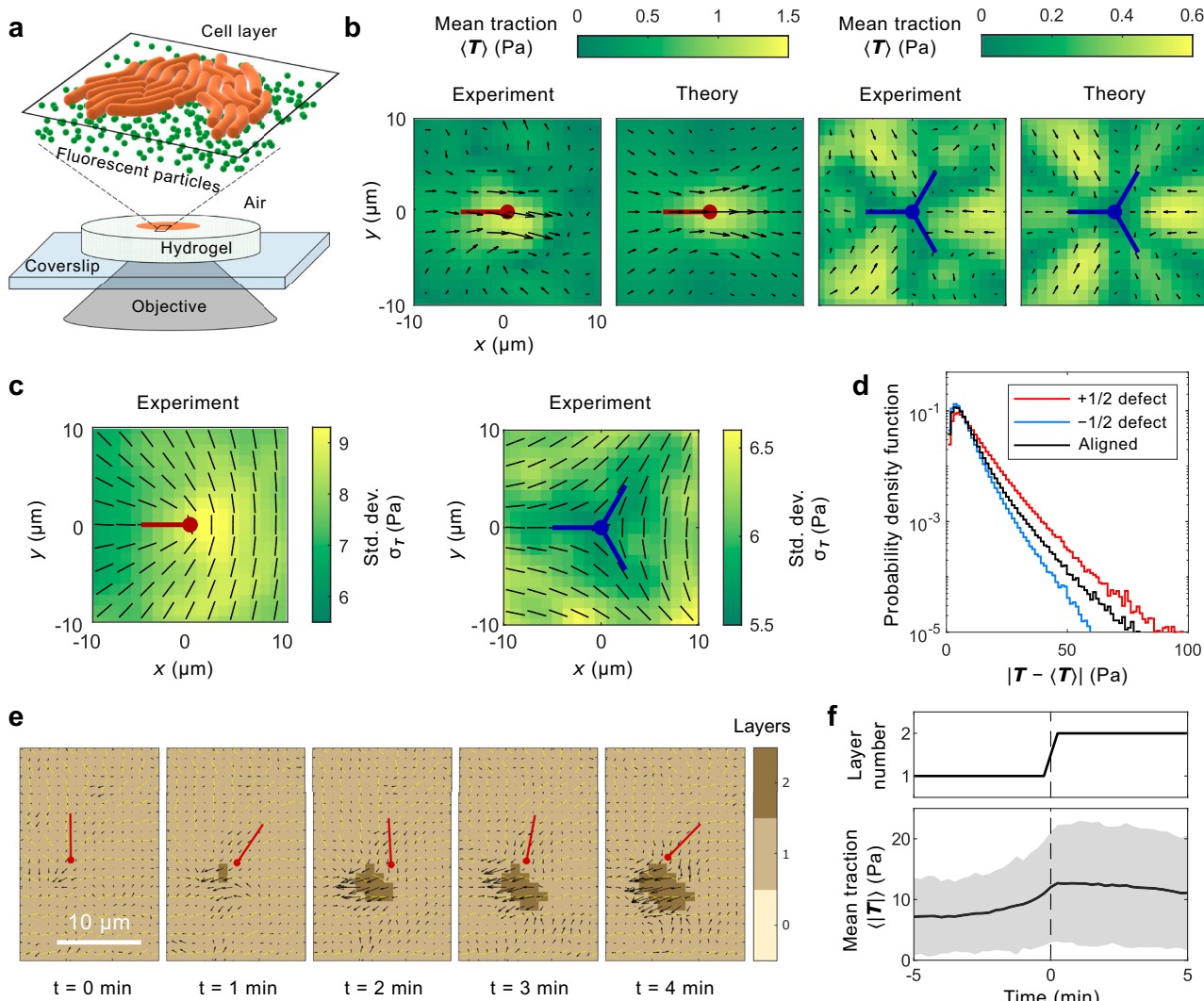

**Fig. 3 | Traction force microscopy (TFM) reveals that large force fluctuations coincide with layer formation. a** Illustration of the TFM setup: *M. xanthus* cells (orange) on a hydrogel surface with embedded fluorescent particles (green) imaged from below. **b** Experimentally measured and theoretically calculated mean traction field ⟨**T**⟩ near ±1/2 defects. The color maps show magnitudes, and the arrows indicate magnitude and direction. **c** Experimentally measured standard deviation of traction $\sigma_T$ near ±1/2 defects. The black lines indicate the mean director field. **d** Distributions of traction fluctuations |**T** − ⟨**T**⟩| within 5 μm from the centers of +1/2 (red) and −1/2 (blue) defects, and in regions where the cells were aligned (black). **e** Traction variation when a second layer formed near a +1/2 defect. The black arrows indicate traction and the yellow lines directors. The white scale bar is 10 μm. **f** Local traction variation when a second cell layer formed around a +1/2 defect. At each location, we shifted the time series of traction magnitude |**T**(t)| such that the formation of the second layer occurred at time $t = 0$ min. Then we calculated its mean (black curve) and standard deviation (gray shade).

there were still consistent, strong traction fluctuations (Fig. 3d). Moreover, traction fluctuations were stronger at +1/2 defects and weaker at −1/2 defects (Fig. 3c, d), although both types of defects had large |∇ · **Q**| near their cores (SI Fig. S5).

Besides the nematic cell-cell active force, the other driving force is the polar cell-substrate active force $\boldsymbol{f}_s^a = \zeta_p \boldsymbol{p}$ that drives gliding motility, where $\zeta_p$ is the polar activity. The fluctuations in these polar active forces will produce fluctuations in the pressure field $P$ in the cell colony. The local build-up of pressure can then trigger the formation of new cell layers. Figure 3e shows an example of a second layer emerging near a +1/2 defect, simultaneously with a strong increase in **T**. As shown in Fig. 2, layer formation events are initiated by strong polarity influx. We identified multiple such monolayer to double-layer transitions and found that the traction magnitude |**T**(t)| increases at the reference time $t = 0$, when the second layer forms (Fig. 3f). The increase in ⟨|**T**(t)|⟩ upon second layer formation was about 5 Pa, which significantly exceeds |⟨**T**⟩| ∼ 1 Pa around +1/2 defects (Fig. 3b). The

traction did not relax immediately after a second layer formed. Similar to $\Phi_p$, |**T**(t)| evolved slowly, on the time scale of minutes (Fig. 3f).

## Cell reversals control local polar order and layer formation

Our results so far show that cell polarity produces strong fluctuations in traction and cell flux, which triggers layer formation. How do cells control local polar order in the system? We now look at how the reversal frequency of *M. xanthus* affects the dynamics of thin colonies. As mentioned above, when the environment turns from nutrient-rich to nutrient-poor, the WT cells prolong their reversal time and increase their speed so that they move for many body lengths before changing direction[28,30,31,39]. To understand the effects of cellular reversal on collective cell motion, traction, and colony morphology, we probed the non-reversing mutant *ΔfrzE* while keeping the nutrient-rich environment invariant, which has been used to simulate the early stage of starvation-induced development of WT or *ΔpilA* cells[28]. Figure 4a shows exemplary maps of layer thickness for reversing and

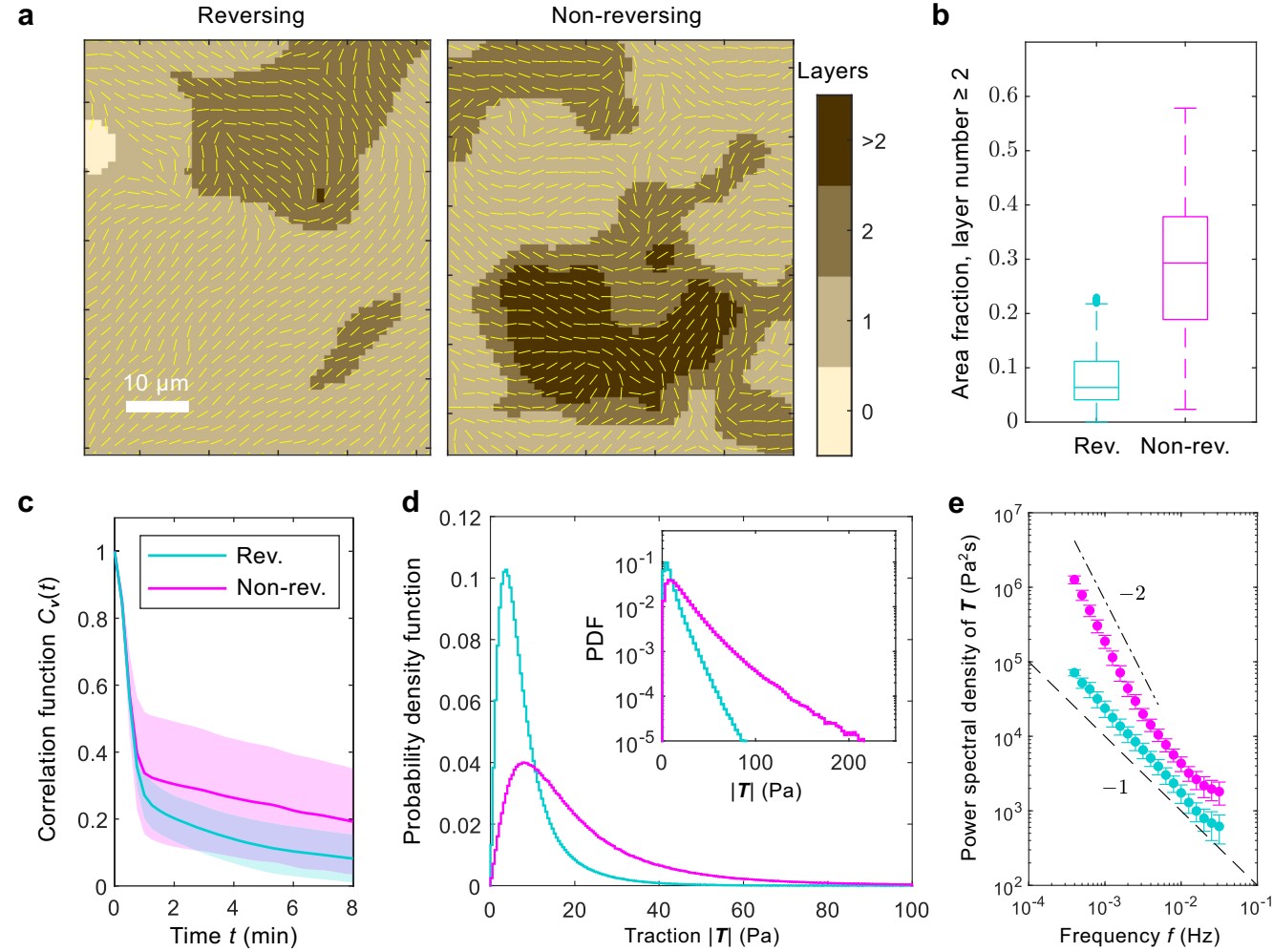

**Fig. 4 | Turning off cell reversal enhances traction fluctuation and layer formation. a** Representative examples of layer-number fields for colonies of reversing (left) and non-reversing (right) cells. Yellow lines indicate the director field $\hat{\boldsymbol{n}}$. The white scale bar is 10 μm. **b** Distributions of area fraction of regions with layer number ≥2 for reversing and non-reversing cells. Center line, median; box limits, upper and lower quartiles; whiskers, 1.5× interquartile range; points, outliers. We measured 10 biological replicates for the reversing cells (WT) and 16 biological replicates for the non-reversing cells (*ΔfrzE*). **c** Temporal auto-correlation functions of velocity $C_v(t)$ (Eq. (8)) for reversing (turquoise) and non-reversing (magenta) cells. The shaded areas show one standard deviation for each cell type. The same colors are used in the following panels comparing these two strains. **d** Distributions of traction magnitude $|\boldsymbol{T}|$. The inset shows the same data in the log-linear scale. **e** Power spectral density of traction. The error bars show one standard deviation. The dashed line has a slope −1 and the dot-dashed line has a slope −2. The data in (**c**–**e**) are from 11 biological replicates for the reversing cells and 13 biological replicates for the non-reversing cells.

non-reversing cells. Across multiple measurements, the non-reversing mutant generated more multi-layer regions than the reversing ones (Fig. 4b). This implies that *M. xanthus* controls $\tau$ in response to starvation to induce layer formation.

Longer reversal time $\tau$ enhances the local polar order in the system. With the polarity assay (see Methods), we measured the temporal autocorrelation functions of $\boldsymbol{p}$ and $\boldsymbol{v}$,

$$C_{\boldsymbol{p}}(t) = \frac{\langle \boldsymbol{p}(t) \cdot \boldsymbol{p}(0) \rangle_s}{\langle |\boldsymbol{p}|^2 \rangle_s} \qquad (7)$$

and

$$C_{\boldsymbol{v}}(t) = \frac{\langle \boldsymbol{v}(t) \cdot \boldsymbol{v}(0) \rangle_s}{\langle |\boldsymbol{v}|^2 \rangle_s}, \qquad (8)$$

respectively, where "$\langle \rangle_s$" represents spatial average. The correlation times of polarity $\tau_{\boldsymbol{p}}$ and velocity $\tau_{\boldsymbol{v}}$ were approximately equivalent. More importantly, they both increased along with the local polarity $p_n$ (see SI Fig. S6). Similarly, we measured $C_{\boldsymbol{v}}(t)$ for reversing and non-

reversing strains (Fig. 4c), and found that the correlation time increased from 0.8 min with reversals to 3.7 min without them, suggesting that local polar order is longer-lived in the non-reversing cell colonies. As a result, although the speeds of reversing and non-reversing cells were similar, the non-reversing populations generated more persistent flows, leading to enhanced aggregation.

As expected, the increased local polar order in the non-reversing cells leads to stronger stress fluctuations, as shown by the longer tail in the traction distribution (Fig. 4d). This increase in traction fluctuations happens predominantly at low frequencies (Fig. 4e). The power spectral densities (PSDs) show two different power laws: traction generated by reversing cells follows a $f^{-1.1}$ power law across almost two decades in frequency $f$, while the PSD for non-reversing cells approaches $f^{-2}$ at low frequencies (Fig. 4e). Understanding the origin of these power laws requires further theoretical investigations.

Lastly, we measured the mean properties of the ±1/2 defects in the *ΔfrzE* colonies (Fig. S7 in the SI). Compared to the reversing cells (Fig. 3 and Fig. S1 in the SI), the mean traction and cell velocity became ~1.5 times larger, and the traction fluctuations became more than 2 times stronger, but no qualitative difference was observed. Although cell

polarity is now very persistent, the colony remains nematic, as shown by the presence of half-integer defects.

## Discussion

Our experiments reveal the coexistence of nematic and polar order in thin *M. xanthus* colonies: The colony behaves as an active nematic system on average, but the instantaneous flows and forces are strongly dominated by the instantaneous polar order due to individual cell polarities. As one cell glides on its neighboring cell, a pair of active forces is generated in the colony, leading to an active nematic stress. In contrast, a cell generates an active polar force in the colony when gliding on a solid substrate. Recent theoretical works showed that fluctuating polar forces in the isotropic phase of a cell layer can lead to nematic order and associated active stresses[40,41]. In contrast, we focused on polar force fluctuations in the nematic phase of our bacterial layers. Furthermore, other theory work studied the flows induced by passive mechanical fluctuations around nematic defects[42]. Instead, we studied active polar fluctuations, and we showed their biological relevance for layer formation.

The strong polarity fluctuations that we observed in *M. xanthus* are due to stochastic cell reversals, and they produce strong traction fluctuations. As shown in Fig. 4, non-reversing cells produce flows with longer temporal correlations as well as stronger traction fluctuations than the reversing strain. Even in colonies of the reversing strain, there are regions with strong and weak local polar order (Fig. 1f), leading to a broad distribution of local traction. Whereas our observations indicate that cell reversals provide an important contribution to traction fluctuations, there can be other intrinsic sources of noise like variability in cellular self-propulsion and cell responses to their local environment. In this sense, living systems may have larger fluctuations than non-living active materials.

As Fig. 3 shows, the average traction fields around ±1/2 defects are well captured by our active nematic model. However, a quantitative model that explains why the traction fluctuations are stronger near the +1/2 defects while weaker near the −1/2 defects does not yet exist. Here, we propose some hypotheses to explain this observation. First, the cell density is higher around +1/2 defects than around −1/2 defects, as they promote the formation of new layers and holes, respectively[14]. Higher cell density may lead to a higher areal density of the gliding motors and hence a stronger active force. Second, anisotropic friction as proposed previously[14] may cause differences between the two defect types. Cells moving in from the tail region of a +1/2 defect are blocked by perpendicularly-oriented cells at the front of the defect. Due to friction being anisotropic, these front cells experience stronger friction, potentially leading to substantial pressure build-up. This pressure could be released whenever cells in front of the defect leave an opening for cells in the defect tail to move into, thus producing high traction fluctuations. These arguments could explain how the nematic order around the defects could control both the average traction field and its fluctuations. Further work on microscopic models of *M. xanthus* colonies may shed light on this issue.

Lastly, most existing active nematic models average over times significantly longer than the characteristic time scale of instantaneous fluctuations, such as those of cell reversal or reorientation. Whereas this averaging approach has successfully captured the dynamics of many systems, our measurements reveal its possible limitations. First, we find that, despite the existence of a characteristic time scale $\tau$ for cell reversal, traction fluctuations can display a scale-free spectrum, as shown by the power laws in Fig. 4e. Therefore, selecting a time scale to average over is not always straightforward. Second, our work shows that short-time fluctuations play an important role for biological functions, such as layer formation, which controls how a colony changes its morphology in response to starvation.

In summary, our experiments reveal stochastic local polar order in thin *M. xanthus* colonies, which not only leads to strong fluctuations in

cell velocity and mechanical stress but also triggers layer formation. Although active nematic theory explains the average flows generated around topological defects and the cell accumulation that promotes layer formation[14], it needs to be extended to capture the large stress fluctuations we measure or the stochasticity in layer formation. Our results show that the polarity fluctuations generated by collectives of self-propelled rod-shaped cells produce stronger forces and flows than the active nematic stresses around topological defects. We further show that *M. xanthus* colonies have found a simple knob—the cell reversal time—that they can use to control internal mechanical stress and colony morphology via tuning the local polar order. This control mechanism enables the colony to spread on a surface when nutrients are abundant and then initiate aggregation when food is scarce.

## Methods

### Preparation of bacterial cultures

In this work, we used four *M. xanthus* strains—WT (DK1622)[14], $\Delta pilA$[38], $\Delta frzE$[43], and $mglB$ :: $mVenus$[35]. To grow the bacteria from frozen stocks, we plated them onto 1.5% agar pads in CTTYE (1% casitone, 10 mM Tris-HCl, 1 mM $KH_2PO_4$, 8 mM $MgSO_4$, and adjusted its pH to 7.6) and kept the plates in a 32 °C incubator for at least 3 days. Then we picked some cells from a colony on the agar plate and transferred them into 10 ml CTTYE solution in a flask to make a liquid culture. The flask was kept at 32 °C overnight with shaking. The optical density (OD) of the liquid culture was measured before each experiment, and we only used cell cultures in the exponential phase.

### Preparation of samples for traction force microscopy (TFM assay)

The samples for traction force microscopy (TFM) experiments were prepared using 35 mm diameter Petri dishes with a 0.17 mm thick glass bottom (Thermo Fisher). The surface of the glass inside the Petri dish was treated with 3-(Trimethoxysilyl)propyl methacrylate (TMSPMA) so that it combined tightly to the polyacrylamide (PAA) hydrogel[44]. The protocol is:

1. Plasma clean the glass surface.
2. Mix 2% (volume fraction) TMSPMA with 98% (volume fraction) 95% ethanol and adjust its pH to 5.0 with glacial acetic acid.
3. Add 1 ml TMSPMA solution in each Petri dish and remove it after soaking for 2 min.
4. Wash each Petri dish three times with pure ethanol and dry it at room temperature for 15 min.

The formulae for making PAA gel at three different stiffnesses are listed in Table S3 in the SI. We always prepared our gel in two steps: first, we made a PAA stock following the formula in Table S4 in the SI, and then we made PAA solution using the corresponding stock. In the main text, all the data were obtained with PAA substrates with a 430 Pa shear modulus. Fluorescent beads (FluoSpheres Carboxylate-Modified Microspheres, yellow-green fluorescent 505/515, 2% solids) with a 110 nm diameter were distributed uniformly in the PAA solution to measure the horizontal and vertical deformations at the gel surface. When making the PAA substrate, we put 15 µl PAA solution in the middle of each Petri dish. Then we put a glass coverslip (Thermo Fisher 12CIR-1, 12 mm diameter) on top of the droplet and waited half an hour for the PAA to gelate. The 12CIR-1 coverslips were treated with water repellent in advance to make their surfaces more hydrophobic.

After gelation, we removed the coverslip and submerged the PAA pad in 66 µg/ml chitosan solution for at least 45 min. The chitosan solution for coating was prepared following these steps:

1. Make 0.2 M Acetic acid (0.12 ml glacial acetic acid to 9.88 ml DI water).
2. Dissolve 10 mg chitosan in 3 ml 0.2 M acetic acid.
3. Make a 1–50 dilution of the chitosan solution prepared in Step 2.

Then we removed the chitosan solution, washed the PAA pad once with sufficient DI water, soaked the pad in sufficient CTTYE for about 10 min, removed the CTTYE, and soaked the pad in fresh CTTYE for another 10 min. After soaking, we removed CTTYE and dried the excess liquid with tissue paper.

To make the cell monolayer, we used overnight liquid cultures of *M. xanthus* in CTTYE shaken at 32 °C. We separated the cells by centrifuging the liquid culture and resuspended them in DI water so that the cell concentration was $8\times$ the concentration of OD = 1 (550 nm wavelength). The cells were dispersed using a pipette for at least 1 min and then shaken on a vortex for at least 30 s. Lastly, we added a 1.5 μl droplet of this concentrated cell suspension onto the surface of the PAA gel, and let it settle for several minutes.

To prevent dehydration of the gel during imaging, we put a 2 mm thick acrylic spacer (cut by a laser cutter) in each Petri dish and put a $22\times22$ mm cover glass on top of the spacer to create a chamber above the substrate and cells. The glass did not touch the gel, so at the air-gel interface, it was gel, cells, and air from bottom up. Outside the chamber, the edges of the spacer and the cover glass were sealed with Valap (1/3 vaseline, 1/3 lanolin, and 1/3 paraffin by mass). After preparation, we kept the sealed samples in a 32 °C incubator for about 2 h, took them out, and kept them all at room temperature during the imaging session.

### Preparation of samples for polarity measurement (Polarity assay)

To measure cell polarity, we used the $mglB::mVenus$ strain, which has fluorescent MglB proteins at the lagging cell pole. Due to the small size of these fluorescent spots, we needed the cell layer to be very flat so that we could capture the polarity of as many cells as possible. As a result, to get images with high enough quality for the analysis, we measured cell polarity using a 1.5% agarose substrate (shear modulus $\approx 50$ kPa), which was much stiffer than the 430 Pa PAA in the TFM assay. Each sample sat on a 25 mm × 75 mm × 1 mm glass slide. We laser cut 1 mm thick rubber spacers with an elliptical hole in the middle and put one on each glass slide. We then dissolved 300 mg agarose powder in 20 ml CTTYE by heating it up, filled each hole with the solution, and put a 22 mm × 40 mm × 0.15 mm cover glass on top to flatten the surface. It took several minutes for the agarose to cool down and solidify. Then we removed the cover glass, and on each pad, put 10 μl liquid culture of $mglB::mVenus$ cells at $2.5\times$ the concentration of OD = 1. We waited several minutes until there was no visible liquid on the gel surface and put a 22 mm × 40 mm × 0.17 mm cover glass on top of the spacer and the gel. We then sealed the edges with Valap and incubated the samples for 2 h at 32 °C.

### Imaging

The images were taken with a commercial Nikon Ti-E microscope with the Perfect Focus System and Yokogawa CSU-21 spinning disk confocal. We used a 60× Plan Apochromat Water Immersion objective (Nikon, NA 1.20, working distance 0.27 mm) and an Andor Zyla camera. The samples were placed in a humidifying chamber, and we kept the temperature 25 °C. For TFM, the objective touched the bottom of the Petri dish and we imaged the gel-cell-air interface from the bottom up, through the substrate. The incident laser beam (488 nm) went through the sample from below while the white light was emitted above the sample. With each sample, we took both time series and $z$-stack images. For the time series, we placed the imaging plane right at the surface of the substrate, so that we saw the cells and the fluorescent particles simultaneously. In each acquisition, we took one bright field image of the cells with white light and one fluorescence image of the fluorescent particles. The time between adjacent acquisitions was 15 s. For the $z$-stack, we moved the focal plane in the $z$ direction (perpendicular to the gel surface) from below the gel surface to above. At each $z$, we took one bright field image and one fluorescence image. In most experiments, the step

size between adjacent $z$ slices was 0.2 μm. In the others, we used a 2 μm step size to scan across a larger range of $z$.

For the polarity measurement, the images were taken through the 0.17 mm thick cover glass. We used white light to take bright field images of the cells and 488 nm laser to image the fluorescently labeled MglB proteins. We define the $z$ position of the imaging plane that cuts through the cell bodies as $z \equiv 0$ μm. In each acquisition, we took one fluorescence image at $z = 0$ μm and three bright field images at $z = -0.9$ μm, 0 μm, and 0.9 μm. The time between adjacent acquisitions was 15 s.

### Data analysis

All experimental data were processed and analyzed using MATLAB (R2019b). For details, see SI Sections IV–VII.

### Reporting summary

Further information on research design is available in the Nature Portfolio Reporting Summary linked to this article.

## Data availability

The raw data underlying the figures in the main text have been deposited on GitHub (https://github.com/endaohan/Data-in-papers). The same data are also provided in the Source Data file. Other data are available from the corresponding authors upon request. Source data are provided with this paper.

## Code availability

The code used for data processing and analysis has been uploaded to GitHub (https://github.com/endaohan/MyxoMonolayer).

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

## Acknowledgements

We thank the Confocal Imaging Facility, a Nikon Center of Excellence, in the Department of Molecular Biology at Princeton University for instrument use and technical advice. We thank Roy Welch and Lotte Sogaard-Andersen for providing *M. xanthus* strains. We thank Matthew Black, Aaron Bourque, Benjamin Bratton, Zemer Gitai, Guannan Liu, Howard Stone, Nan Xue, Cassidy Yang, and Rui Zhang for their assistance in the research and many useful discussions. This work was supported by the NSF through awards PHY-1806501 and PHY-2210346, and the Center for the Physics of Biological Function (PHY-1734030). E.H. acknowledges the NTU Start-Up Grant. R.A. acknowledges funding from the European Union through the ERC Starting Grant "Living Fluctuations" (No. 101114584). N.S.W. acknowledges National Institutes of Health Grant R01 GM082938.

## Author contributions

E.H. and J.W.S. conceived the research. E.H. performed the experiments and data analysis. C.F. and R.A. derived the theoretical model. K.C. and M.D.K. provided key data processing and experimental techniques, respectively. N.S.W. and J.W.S. supervised the project. All authors designed the research and interpreted the results. E.H., C.F., R.A., N.S.W., and J.W.S. wrote the paper with input from the other authors.

## Competing interests

The authors declare no competing interests.
