## [Transparent Peer Review file · Nature Communications]

Local polar order controls mechanical stress and triggers layer formation in *Myxococcus xanthus* colonies

Corresponding Author: Professor Endao Han

Version 0:

Reviewer comments:

Reviewer #1

(Remarks to the Author)

Review Report of paper titled "Local polar order controls mechanical stress and triggers layer formation in developing *Myxococcus xanthus* colonies"

In this work authors performed the experiments to show the effect of local polar order on the morphology of the *Myxococcus xanthus* colonies. It is found the local polar order controls the mechanical stress and trigger the layer formation in the colony. The biological path ways that controls the decision to form a fruiting body have been studied extensively. But the mechanical trigger which controls the formation of multilayers in colonies is rarely studied previously. In that direction this paper gives a new mechanism to understand the formation of layers in cell colonies. Which can have lots of potential applications to control and artificially designed the system. Hence I find the work is relevant and is an important addition to understand the morphology of cell colonies in the terms of mechanical stress and force.

But before the work can be published in the journal I suggest the following modification to improve the quality of the manuscript.

1. I find that the introduction is written very briefly and most of the text and citations till the second paragraph have discussed the general properties of polar order in collective behaviour of self propelled particle systems. But I am sure there must be many other study either experiments, theory or simulation, which might have studied the formation of multilayers in colonies of self propelled particles. It might be possible that the explanation of such layer formation is unexplored in detail. It is better if authors can include few such work in the introduction and discuss about them in brief and how the work presented here answers the questions unanswered there.
2. I find equation 1 on page 1, is important, but at the end of the manuscript the sign of the force term is discussed for the contractile and extensile case. It is better to write about it immediately after that equation and mention the sign of the force for the current system.
3. On page 3 first paragraph, last line it is mentioned that the "Our results now show that this average", it is better if the authors can show the influence of cell polarity at some other places (other work), where it has been more important than the other local fields. Also it is required to give an explanation (more microscopic) why the cell polarity is more important compare to other local field.
4. On the same page second column last paragraph "Instead of accumulating cells at a steady state.....". It is required to explain how this conclusion is derived and why it should happen. From the given explanations in the text it is not clear.
5. In the same paragraph, the state is made about the stochasticity in the system, I find again this statement is vague and need some more explanation or some additional plot to show it.
6. What is traction force. Although it is given in detail on the page 18th of the SM, but it is required to define it in brief in the main text also.

7. It is good to provide more microscopic understanding why the traction force will be stronger near the core of the +1/2 defect.

8. At the last page, the results are shown in terms of cell reversal time.

I feel the cell reversal time should also depend on the density of cell. It is good if the authors can provide some idea of reversal time on the packing density and can comment that in what range of packing density their results are valid.

9. It is required to write the discussed in more detail, right now it is very brief.

Reviewer #2

(Remarks to the Author)

In this paper, Han et al. investigate layer formation in *Myxococcus xanthus* colonies. To this end, they combine detailed measurements of cell orientation, velocity, polarity, and force at the single-cell level to reveal the presence of a stochastic local polar order. Notably, the results indicate that this local polar order, in addition to the well-known nematic order, plays a significant role in controlling mechanical stress and layer formation within the colonies. The paper is well-written, and despite the incompleteness of the theoretical model, the strength of the experimental results and the significance of illustrating the intriguing interplay of multiple orientational orders in orchestrating a morphological transition make this manuscript suitable for publication. I have some minor comments:

1) Perhaps the intriguing experimental observation that fluctuating polar forces can induce large fluctuations in the active force is related to theoretical models where fluctuating polar forces can induce extensile active stresses, see for example F. Vafa, M. Bowick, B. Shraiman, C. Marchetti, *Soft Matter* 2021

L. Bonn, A. Ardaseva, R. Mueller, T. Shendruk, A. Doostmohammadi, *PRE* 2022

A. Killeen, T. Bertrand, C. Lee, *PRL* 2022

2) In the main text, it would be helpful to clarify exactly what kind of average $\langle \cdot \rangle$ means.

3) How good are the various fits, e.g., what are the residuals?

4) In comparing Eq. S19 to Eq. S18a, why is it the case that $\langle Q \cdot v \rangle = \langle Q \rangle \cdot \langle v \rangle$? I would have imagined Q and v to not be independent.

5) In Eqs. S19-21, what is p ? Relatedly, in Eq. S22, what is \tilde{P} ?

6) In the middle of the left column of p. 5, "reversal" is misspelled.

Reviewer #3

(Remarks to the Author)

The manuscript "Local polar order controls mechanical stress and triggers layer formation in developing *Myxococcus xanthus* colonies" combines careful biophysical measurements and theoretical models to extend their recent work (Ref. [15]) that explains the formation of multilayer aggregate based on the nematic orientation defects. The authors found an important role of "a stochastic local polar order" that results in large fluctuations of both cell velocity and active force at topological defects. Overall, I found the measurements performed in this paper very interesting. The conclusions about the importance of polar order are well-supported by the data presented but somewhat unsurprising. Indeed, given the typical length-scale of the defect \sim several cell lengths and the time-scale of the second layer formation as reported here and in Ref. 15 do not allow to "average" over the polar motion of individual cells (cells several body lengths before switching the polarity). It is more surprising why the mean behavior still agrees with the nematic theory developed in Ref. 15 but this question seems left unanswered. The authors also aim to connect the observations with the developmental triggering of aggregation in station conditions. These connections between the observations and starvation-induced development remain purely hypothetical and not supported by the reported data.

General comments:

1. Strains used: The authors selected multiple strains for different measurements. This justification is buried inside of SI and does not seem convincing. First of all, "to simplify the intercellular and cell-substrate interactions" the authors used a Δ pilA mutant, which does not make any type IV pili and thus has no S-motility (A+S $^-$). This mutant, however, in addition to lacking S-motility this strain also produces very little extracellular matrix (ECM) which not only affects cell motility but also serves as a signaling molecule (dif signaling). This could be simply a limitation of the generality of the study, i.e. authors could just focus on aggregation in that background, but other strains that were used for particular measurements (e.g. cell tracking and polarity detection were not in the same background), e.g. *mgIB::mVenus* strain with fluorescent label on the lagging pole was A+S $^+$. The non-reversal mutant used was also in A+S $^+$.

2. Manuscript structure: As written the paper combines the results and discussion and therefore it is hard to separate authors' conclusions based on their data, speculations/interpretations, and something that is supported by the literature. Despite the large number of cited papers, it is not clear whether the cited sources support the claim; in some cases, very general reviews or several are cited for very specific claims. The authors are advised to trim their reference list and use primary literature. The SI is very loosely connected to the main text – better integration will help to strengthen the paper.

3. The first major conclusion that the authors reach via quantification of polarity and averaged local velocity is that cell flow is mainly driven by instantaneous polarity. This conclusion is achieved from the observed correlation between "local cell velocity" and "local polarity". However, I see this correlation to be a straightforward consequence of the averaging procedure employed to compute "local cell velocity". Authors use this argument themselves with the cartoon insets in Fig. 4f. I think due to the averaging procedure the implied term "local cell velocity" is misleading and the "cell flow" chosen in the title or cell net

flux is more appropriate. I think a somewhat less trivial finding is that the presence of defect slightly changes the relationship between “local cell velocity” and “local polarity”. However, this is again can be deduced from the observations that orientation defects do not significantly slow down individual cells. If the authors agree with this simple averaging interpretation they should probably say that explicitly in this section. If the effect is stronger or weaker than simple averaging interpretation suggests that should be explicitly stated.

4. In the same way, I find the results presented in Fig. 2 a straightforward consequence of averaging and conservation laws but they nevertheless serve as an important control in data processing.

5. I found the results of traction force microscopy measurements the most novel and interesting aspects of the work. Interestingly, the nematic theory can predict the average traction forces despite the huge std deviations. Can authors comment on why this is the case? Can the theory say something about the fluctuations? What is the major source of the fluctuations – defect-to-defect differences? Or stochasticity of cell behaviors? It may be worth discussing in the main text how the theory used here differs from that presented in Ref. 15.

6. Again, from the averaging perspective higher polar order in cells that do not reverse is not too surprising. However, I found the author’s attempt to connect the results to the increased polar order of the system with starvation-induced aggregation unconvincing. The importance of reversals on the collective cell alignment and motility patterns has been demonstrated in the earlier publications of Shaetitz lab (<https://doi.org/10.1098/rsif.2015.0049>) and by others (<http://dx.doi.org/10.1098/rsfs.2012.0034>). These results demonstrate that nearly complete elimination of the reversals in $\Delta frzE$ would be quite distinct from a relatively small (on average) and highly heterogeneous increase in reversal times at the onset of the developmental aggregation. Moreover, under non-starvation conditions, non-reversal strains can persist for a long time without forming fruiting-body-like aggregates indicating that the formation of the multiple layers observed in the defects and formation of fruiting bodies may not be controlled by the same mechanism. Perhaps a better way to assess how moderate perturbations in average reversal frequency affect the polar order and layer formation would be to use chemical attracts or repellants that are known to modulate the reversal period of wild-type cells (casitone, isoamyl alcohol etc. see e.g. <https://doi.org/10.1046/j.1365-2958.1997.4261783.x>)

7. In light of the above and given that none of the experiments were performed under starvation-induced development, I would strongly suggest removing “developing” from the title and other places in the manuscript. Conclusion “we show that polarity fluctuations trigger the formation of new cell layers, which enables the starvation-induced development from monolayers to droplet-like fruiting bodies in the social bacterium *Myxococcus xanthus*” advertised in the intro is similarly unjustified.

Version 1:

Reviewer comments:

Reviewer #1

(Remarks to the Author)

I find the authors have addressed the most of the comments raised by the referee's and also made the required modification in the manuscript. Hence now I feel manuscript is suitable for publication in Nat. Comm.

(Remarks on code availability)

I find the authors have addressed the most of the comments raised by the referee's and also made the required modification in the manuscript. Hence now I feel manuscript is suitable for publication in Nat. Comm.

Reviewer #2

(Remarks to the Author)

I thank the authors for their response. Their clarifications and revisions have addressed my comments and I recommend publication of the manuscript.

(Remarks on code availability)

Reviewer #3

(Remarks to the Author)

After reading the author’s response letter and revisions, I still have three points concerning the results and interpretation. 1) Regarding the fundamental nature of the active nematic and the self-propelled rod models in application to the description of the myxobacteria: In my interpretation, the active nematic model is fundamentally phenomenological (hydrodynamic description). It can be formulated by the symmetry arguments based on the postulated energy functional or derived from the microscopic description under certain limits and assumptions (<https://arxiv.org/html/2401.05070v1>). Indeed, in the previous manuscript by some of the authors (Ref. 14), it was shown that their data on the average flow fields around defects is consistent with a particular form of the active nematic model. However, no microscopic justification for this has been provided. The authors answer my question on this with a somewhat superficial ‘averaging’ argument. However, this averaging is not trivial, especially in a system with strong and non-linear interactions, and when the time/length scales of the mesoscopic phenomena (defect) and the microscopic phenomena (free path of a cell) are of the same order. On the microscopic scale, unlike the microtubule system, the authors reference in their response or the preprint above, myxobacteria cells are polar. Therefore, the self-propelled rod model is a more appropriate mechanistic behavior. Thus, I still

think that interpreting the importance of the polar order based on the breakdown of the averaging assumptions required is the most obvious interpretation of the results and is totally expected. From this perspective, the averaging is even more likely to fail if agents stop switching their polarity. From the same perspective, the equations and theories presented in the main text need to be explained/defined with more rigor. For example, it is not clear if the force balance in Eq. (4) represents a force balance on the microscopic scale (single cell) or mesoscopic (hydrodynamic small volume with many cells).

To summarize, given the fundamentally polar nature of force generation in moving cells, it is by no means surprising that polar effects are significant and lead to large fluctuations from the active nematic description. The authors claim these fluctuations are “anomalously” large, but their “anomalous” nature has not been demonstrated. It is even less surprising that polar effects are more important when cells do not reverse. What is surprising is that at topological defects, the average cell velocity (result in Ref. 14) and active force (new here) agree with predictions from active nematic theory. But that effect remains unexplained.

2) I find data on Fig. iii in the response very interesting and recommend its inclusion into the manuscript. It does not fully alleviate my concern about changing from strain to strain in different measurements but perhaps supports it further. I cannot fully follow the theoretical arguments presented. Suppose the force balance equation (4) is interpreted in a hydrodynamic sense. In that case, the pili forces between cells in that averaging volume should vanish or be absorbed into a negative pressure term, and the forces from pili extending away from the volume should, on average, be in the direction of the nematic alignment. And given that traction forces for WT cells do not completely agree with the active nematic model (is it?), it is surprising that in Ref. 14 for WT, average cell velocity nevertheless matches that theory.

3) The authors continue (but to a lesser extent) to imply that their results are relevant for the multi-layer formation in starvation conditions. For example, in the intro, the paragraph starting with “In this work” continues with the sentence, “As nutrients become scarce, the cells increase τ and start to form double or triple layers on top of the original monolayer.” This has not been demonstrated here, and the causal effects of these phenomena remain questionable. It should be noted that (1) the changes in reversal frequency during starvation often happen in the later stages of development after the initial “cell towers” have been established; (2) Development often experimentally observed at cell densities that implied initially multi-layer situation; (3) The extent of changing the reversal frequency cited is not consistent with other pubs (<https://doi.org/10.1128/jb.00544-07>, <https://doi.org/10.1073/pnas.1620981114>)

To be clear – these concerns are mainly not about the data presented but about its interpretation, so I don’t want my feedback to stand in the way of paper publication.

(Remarks on code availability)

Version 2:

Reviewer comments:

Reviewer #3

(Remarks to the Author)

The authors adequately addressed my previous comments.

(Remarks on code availability)

Reply to the reviewers' comments on **Local polar order controls mechanical stress and triggers layer formation in developing *Myxococcus xanthus* colonies**

Endao Han, Chenyi Fei, Ricard Alert, Katherine Copenhagen, Matthias D. Koch, Ned S. Wingreen, and Joshua W. Shaevitz

We thank all the reviewers for carefully reading our manuscript and providing many inspiring and constructive suggestions. We have answered the questions raised one by one and revised the manuscript accordingly. The original comments by the reviewers are in **blue**, and our responses are in **black**. The revisions in the original manuscript are highlighted in **blue**. The figures in this reply are labelled as Fig. i, Fig. ii, and so on to avoid confusion with the figures in the manuscript, which are labelled as Fig. 1, Fig. 2, etc.

Reviewer #1 (Remarks to the Author):

Review Report of paper titled "Local polar order controls mechanical stress and triggers layer formation in developing *Myxococcus xanthus* colonies"

In this work authors performed the experiments to show the effect of local polar order on the morphology of the *Myxococcus xanthus* colonies. It is found the local polar order controls the mechanical stress and trigger the layer formation in the colony. The biological path ways that controls the decision to form a fruiting body have been studied extensively. But the mechanical trigger which controls the formation of multilayers in colonies is rarely studied previously. In that direction this paper gives a new mechanism to understand the formation of layers in cell colonies. Which can have lots of potential applications to control and artificially designed the system. Hence I find the work is relevant and is an important addition to understand the morphology of cell colonies in the terms if mechanical stress and force.

But before the work can be published in the journal I suggest the following modification to improve the quality of the manuscript.

1. I find that the introduction is written very briefly and most of the text and citations till the second paragraph have discussed the general properties of polar order in collective behaviour of self propelled particle systems. But I am sure there must be many other study either experiments, theory or simulation, which might have studied the formation of multilayers in colonies of self propelled particles. It might be possible that the explanation of such layer formation is unexplored in detail. It is better of authors can include few such work in the introduction and discuss about them in brief and how the work presented here answers the questions unanswered there.

R: Surprisingly, we did not find much other work on layer formation in systems of self-propelled particles like ours. We found four papers with a relatively close relationship to what we report in this manuscript:

- S.C. Takatori and K.K. Mandadapu, arXiv preprint arXiv:2003.05618: A combined experimental and theoretical study of self-propelled rod-like bacteria forming double layers, explained using a model of self-propelled particles, for which the control parameter is the Peclet number. However, it did not focus on the nematic and polar order in the system.
- A. Sciortino and A.R. Bausch, PNAS, 118(6):e2017047118, 2021: A study of propelled actin filaments with both polarity and nematic order. Actin concentration increases at +1/2 defects, reminiscent of layer formation in our system. No quantitative model is provided.

- Starruß *et al.* Interface Focus, 2:774, 2012: This work uses a self-propelled rod model to explain aggregation of *M. xanthus*. However, it mostly focuses on how the bacteria move on a bare surface and form clusters of different sizes. The formation of multiple layers is mentioned, but not studied in any detail. Furthermore, the model provided is a kinetic model that does not account for mechanical forces.
- O. J. Meacock, *et al.* Nature Physics, 17(2):205–210, 2020: A study of self-propelled rod-like bacteria. Collisions of two +1/2 defects lead to verticalization of the bacteria instead of layer formation. Interestingly, the authors performed numerical calculations with a self-propelled rod model but analytical calculations with an active nematic model, and provide identical results. This is the opposite of what we observe in our system, where we find that the polar fluctuations around the average nematic order drive layer formation.

We have added a discussion of these papers to the new introduction, highlighting why our system and perspective is different from previous work.

2. I find equation 1 on page 1, in important, but at the end of the manuscript the sign of the force term is discussed for the contractile and extensile case. It is better to write about it immediately after that equation and mention the sign of the force for the current system.

R: As suggested, we have fixed this problem by adding a sentence explaining the sign of the active force right after Eqn. 1, which reads “The active stress along the directors is extensile if $\zeta_{\text{n}} > 0$ and contractile if $\zeta_{\text{n}} < 0$.” We have also revised related sentences in the section “Cellular tractions and layer formation” accordingly to avoid repetition.

3. On page 3 first paragraph, last line it is mentioned that the “Our results now show that this average”, it is better if the authors can show the influence of cell polarity at some other places (other work), where it has been more important than the other local fields. Also it is required to give an explanation (more microscopic) why the cell polarity is more important compare to other local field.

R: Combining the suggestions of all three reviewers, in the revised manuscript, we further addressed this issue in the introduction and in a new Discussion section. In the revised manuscript, we briefly review the consequences of polarity fluctuations in active systems, such as for layer formation in the Introduction (this is also related to the Reviewer’s comment #1), and we discuss the links between nematic force, polar force, and fluctuations in the Discussion.

We do not currently have a quantitative model to explain why polarity is more important than the other fields or why the polar active force is stronger than the nematic active force in our system. This is also related to Reviewer #2’s comment 1. At this stage, it is an experimental observation. Our guess, however, is that polar fluctuations are particularly strong in our system due to the sudden polarity reversal events in Myxo cells. We revised the last sentence of the paragraph the Reviewer referred to read, “Our results show that this average velocity due to the nematic active force $\langle \mathbf{f}^{\text{a}}_{\text{n}} \rangle$ is small compared to the velocities produced by polarity fluctuations (Fig. 1f). These strong polarity fluctuations could be due to the sudden polarity reversals of *M. xanthus* cells shown in Fig.~\ref{fig:polarity}e.” We also provide some discussion of this topic in the newly added Discussion section. Please refer to the revised manuscript for details.

4. On the same page second column last paragraph “Instead of accumulating cells at a steady state.....”. It is required to explain how this conclusion is derived and why it should happen. From the given explanations in the text it is not clear.

R: The former understanding of layer formation near a +1/2 defect was that the cell flow is caused by the nematic order in the cell layer. It is driven by the nematic active force and balanced by the anisotropic cell-substrate friction. The active force is the strongest near the core of the +1/2 defect pointing toward its head, and the anisotropic friction is stronger in the direction perpendicular to the cells and weaker along the cells. As a result, the cell velocity ahead of the defect core is slower than that moving inward near the tail of the defect. To illustrate this effect, we added an extra panel in Fig. 2 (new Fig. 2a). This asymmetric flow causes a net influx of the cells and thus the formation of a new layer. The experimentally measured average velocity and traction both match this predicted pattern very well, which means that on average, the cell flow behaves as predicted for an active nematic.

This picture of the average behavior suggests that, once layer formation has started, the cells near the defect core should be extruded at a constant rate, and a second layer should be built up continuously and steadily. However, here we take a step further by looking at how the second layer initially appears. We show that polarity fluctuations produce strong flow fluctuations around defects, which correlate with the onset of layer formation (current Fig. 2c-e). We have therefore revealed the fluctuations that drive layer nucleation, which is a fast discontinuous process.

To make this point clearer in the revised manuscript, we made two changes to Fig. 2:

- (1) We added a new panel Fig. 2a to illustrate the idea of cell flow driven by nematic order and by cell polarity.
- (2) We added a black dot-dashed straight line in the current Fig. 2d that shows the average cell accumulation speed estimated using the mean velocity field near the +1/2 defects.

In the text, we made multiple changes in the section *Polarity-driven cell influx drives layer formation*. Please refer to the manuscript for the detailed revision.

5. In the same paragraph, the state is made about the stochasticity in the system, I find again this statement is vague and need some more explanation or some additional plot to show it.

R: We thank the reviewer for raising this point, and we apologize for having used the term stochasticity in an unclear way. What we tried to explain there is why in the current Fig. 2d (previous Fig. 2c), the average polarity-based flux shows a consistent negative value when a second layer forms, while the average velocity-based flux is not as consistent: it fluctuates more and even jumps to zero at some points. Our argument is that the cell velocity is strongly affected by the cell polarity, but there are other influences as well, such as forces from neighboring cells. These factors lead to fluctuations in the velocity field that do not directly correlate with those of the polarity field, thus causing the slightly different behaviors in the velocity and polarity fluxes. We have revised the text to clarify this point.

To make this point clearer, we added one more panel in Fig. 2 (new Fig. 2e). We show the distributions of velocity-based and polarity-based fluxes both when a double layer formed (purple) and when it did not form (green). The two peaks are more separated for the polarity flux Φ_p

compared to the velocity flux Φ_v . We have also revised the last paragraph of this section correspondingly. Please see the revised manuscript for details.

6. What is traction force. Although it is given in detail on the page 18th of the SM, but it is required to define it in brief in the main text also.

R: We have revised the first two paragraphs of the section *Cellular tractions and layer formation* based on the Reviewer's suggestion. We have moved some equations and explanations in the SM to this paragraph so the readers can better understand the physical meaning of the measured traction force. Please see the revised manuscript for details. We also rearranged the panels in Fig. 3 for better presentation.

7. It is good to provide more microscopic understanding why the traction force will be stronger near the core of the +1/2 defect.

R: For the average traction near the core of the +1/2 defect, our interpretation is that this reflects the nematic active force. At the defect core, there is stronger deformation in the nematic order, thus the nematic active force is stronger.

For the fluctuations in the traction force, we do not have a comprehensive quantitative model yet, but we speculated to rationalize the observation:

(1) Higher cell density, including higher density within a monolayer or the formation of double or triple layers. Higher cell density leads to higher density of gliding motors, and thus stronger active force. Since +1/2 defects promote cell aggregation while -1/2 defects promote cell depletion, the resulting changes in cell density might enhance or reduce traction fluctuations around +1/2 and -1/2 defects, respectively.

(2) Anisotropic friction. Related to the nematic order, the cells moving in from the comet tail region of a +1/2 defect are blocked by the perpendicularly-oriented cells in front. Because of friction anisotropy, the front cells experience high friction, and hence substantial pressure can build up at +1/2 defects, potentially leading to higher traction fluctuations.

We have added these points to the Discussion section.

8. At the last page, the results are shown in terms of cell reversal time. I feel the cell reversal time should also depends on the density of cell. It is good if the authors can provide some idea of reversal time on the packing density and can comment that in what range of packing density their results are valid.

R: We thank the reviewer for this question. For the strain of *M. xanthus* that we use (DK1622), our group has previously shown that the reversal time is largely independent of cell density. We directly measured and compared the reversal frequencies of cells inside a thin colony to lone cells moving on an open substrate. The results were reported in the Supplementary Information of the paper Liu et al. PRL 122(24) 248102 (2019) and Guannan Liu's PhD thesis. We copied a figure from the thesis and show it below (Fig. i). Wild-type *M. xanthus* cells in groups have a slightly higher reversal frequency than the individual ones, but based on the *p* value ($p=0.1$) given by the Kolmogorov-Smirnov test, the difference is not significant. We note that for another WT strain,

Figure i. Distribution of reversal frequencies for wild-type cells in groups at high density (blue) and in isolation at low density (red). Figure from Liu's PhD thesis – *The mechanical basis of Myxococcus xanthus self-organization and motility: from single cells to collective behavior* (2019).

DZ2, there are experimental measurements that support both density dependence and independence of the reversal time. Whereas Wu et al. PNAS 106(4) 1222-1227, 2009 found no significant difference in reversal frequency between low-density and high-density cell groups, an earlier paper by Shi et al. Proc. Natl. Acad. Sci. USA 93 4142-4146, 1996 found that the reversal frequency decreases as the cell density increases. They attributed this decrease to the cell-cell interactions mediated by pili. Using a mutant without pilus (A+S-), the reversal frequency became independent of the cell density.

Based on these results, we hypothesize that in our experiments with thin colonies of WT DK1622 (A+S+) and a pilus-deficient mutant of this strain, $\Delta pilA$ (A+S-), the cells' reversal frequency is approximately independent of the cell density. In the manuscript, we have added “ ..., and its reversal time τ is approximately independent of the cell density in the colony.” in the paragraph about cell reversal, which was in the section *Cell reversals control local polar order and layer formation* in the old version, but has been moved to the Introduction in the revised manuscript.

9. It is required to write the discussed in more detail, right now it is very brief.

R: As the reviewer suggested, we have added a new Discussion section to sum up our results and what we learn from them. In it, we briefly summarized the observations from our experiments and their implications, such as strong velocity and traction fluctuations, coexistence of nematic order and polarity fluctuations, the relationship between polarity fluctuation and layer formation, and the effects of cell reversal frequency. We also addressed some of the other reviewers' suggestions and questions in the new Discussion section.

Reviewer #2 (Remarks to the Author):

In this paper, Han et al. investigate layer formation in *Myxococcus xanthus* colonies. To this end, they combine detailed measurements of cell orientation, velocity, polarity, and force at the single-cell level to reveal the presence of a stochastic local polar order. Notably, the results indicate that this local polar order, in addition to the well-known nematic order, plays a significant role in controlling mechanical stress and layer formation within the colonies. The paper is well-written, and despite the incompleteness of the theoretical model, the strength of the experimental results and the significance of illustrating the intriguing interplay of multiple orientational order in orchestrating a morphological transition make this manuscript suitable for publication. I have some minor comments:

R: We thank the reviewer for these encouraging words.

1) Perhaps the intriguing experimental observation that fluctuating polar forces can induce large fluctuations in the active force is related to theoretical models where fluctuating polar forces can induce extensile active stresses, see for example

F. Vafa, M. Bowick, B. Shraiman, C. Marchetti, *Soft Matter* 2021

L. Bonn, A. Ardaseva, R. Mueller, T. Shendruk, A. Doostmohammadi, *PRE* 2022

A. Killeen, T. Bertrand, C. Lee, *PRL* 2022

R: We have read these three papers carefully and indeed they are very inspiring. We have added some discussions about these theoretical works in the new Discussion section. We do see two major differences between these theoretical predictions and our system.

The first major difference is that we focus on polar fluctuations in the nematic phase, and we show their biological significance for layer formation. Instead, the papers by Vafa et al. and Killeen et al. focus on polar fluctuations in the isotropic phase, and show that they provide a mechanism to generate nematic order. Furthermore, the paper by Bonn et al. studies passive mechanical fluctuations around nematic defects, whereas we focus on active polar fluctuations.

Second, in all three papers, a separation of time scales is assumed. The authors assume that there is a characteristic time scale that represents the instantaneous fluctuations, and they predict the emerging average order on a much longer time scale. However, our measurements show two different features in our system that reveal the limitations of such an approach: (a) The short-time dynamics play an important role in controlling layer formation and hence the three-dimensional morphology of the colony, and (b) Even though the system has an intrinsic time scale, which in our case is the reversal frequency, we find that the frequency spectrum of traction fluctuations is scale-free, as shown by the power laws in Fig. 4e. Therefore, selecting a time scale to average over is not always straightforward.

Please refer to the Discussion section of the updated manuscript for the detailed revisions.

2) In the main text, it would be helpful to clarify exactly what kind of average $\langle \cdot \rangle$ means.

R: In our paper, there are four different types of averages. Previously, we used the same symbol $\langle \cdot \rangle$, which could cause confusion as the Reviewer pointed out. To address this issue, now we use four different notations:

- For the average in the comoving frame of the defects, we keep the symbol $\langle \cdot \rangle$. This applies to the mean velocity, traction, directors, and other properties around the defects.

- For the sample average of the data from different experiments, we use an overline. This applies to the average fluxes in the section *Polarity-driven cell influx drives layer formation*, the average correlation time, and so on.
- To obtain the average temporal correlation functions $C_v(t)$ and $C_p(t)$ taken from different locations or the average velocity or polarity in a square region, spatial averages were calculated. We define the spatial average using the symbol $\langle \rangle_s$, where “s” represents “spatial”.
- In the Section IV of the SI, we used $\langle \rangle_R$ to represent the average within a circle with radius R. The meaning of this symbol has been described in the text.

Multiple corresponding changes have been made in the main text, SI, and the figures. Now the meanings of these notations are all defined the first time they appear in the manuscript. Please refer to the manuscript for the specific revisions.

3) How good are the various fits, e.g., what are the residuals?

R: In Fig. 3b, we fitted the theoretical predictions to the experimental traction fields near the +1/2 and -1/2 defects. The differences between the experimental and theoretical traction and cell velocity are now shown in Fig. S1.

In the Supplemental Information, we also fit the order parameter profile and the velocity flux around defects. The results are shown in Fig. S8c and Fig. S10b. The corresponding residual plots have been added to these figures.

4) In comparing Eq. S19 to Eq. S18a, why is it the case that $\langle Q \cdot v \rangle = \langle Q \rangle \cdot \langle v \rangle$? I would have imagined Q and v to not be independent.

R: We assume that the fluctuations in Q and v are uncorrelated, which leads to $\langle (Q - \langle Q \rangle) \cdot (v - \langle v \rangle) \rangle = \langle Q \cdot v \rangle - \langle Q \rangle \cdot \langle v \rangle = 0$. The reviewer is right in that velocity and nematic are related through force balance, and hence they are not fully independent. However, in our system, a cell reversal event causes a velocity fluctuation without affecting the nematic order. We think that cell reversals are the main source of velocity fluctuations, and hence that our approximation is reasonable.

To further test this assumption, we calculated $\langle Q \cdot v \rangle$ and $\langle Q \rangle \cdot \langle v \rangle$ separately and compared them. In Fig. ii below we show the x and y components of $\langle Q \cdot v \rangle$, $\langle Q \rangle \cdot \langle v \rangle$, and $\langle Q \cdot v \rangle - \langle Q \rangle \cdot \langle v \rangle$ around defects. The difference is much smaller than either term, confirming that $\langle Q \cdot v \rangle$ and $\langle Q \rangle \cdot \langle v \rangle$ are approximately equal.

We have added these arguments and Fig. ii into the SI of the manuscript.

5) In Eqs. S19-21, what is p? Relatedly, in Eq. S22, what is tilde P?

R: We have corrected the typos in these equations. In the manuscript, we use \mathbf{p} to represent the polarity field, P to represent the pressure in the cell colony, and \tilde{P} is the Fourier transform of P . In some of these equations, we used p where it should have been P . We have now fixed this mistake.

Figure ii. Compare $\langle \mathbf{Q} \cdot \mathbf{v} \rangle$ and $\langle \mathbf{Q} \rangle \cdot \langle \mathbf{v} \rangle$. Their difference is much smaller than individual term, so these two terms are approximately equal.

6) In the middle of the left column of p. 5, “reversal” is misspelled.

R: Thank you, we have corrected the spelling.

Reviewer #3 (Remarks to the Author):

The manuscript “Local polar order controls mechanical stress and triggers layer formation in developing *Myxococcus xanthus* colonies” combines careful biophysical measurements and theoretical models to extend their recent work (Ref. [15]) that explains the formation of multilayer aggregate based on the nematic orientation defects. The authors found an important role of “a stochastic local polar order” that results in large fluctuations of both cell velocity and active force at topological defects. Overall, I found the measurements performed in this paper very interesting. The conclusions about the importance of polar order are well-supported by the data presented but somewhat unsurprising. Indeed, given the typical length-scale of the defect \sim several cell lengths and the time-scale of the second layer formation as reported here and in Ref. 15 do not allow to “average” over the polar motion of individual cells (cells several body lengths before switching the polarity). It is more surprising why the mean behavior still agrees with the nematic theory developed in Ref. 15 but this question seems left unanswered. The authors also aim to connect the observations with the developmental triggering of aggregation in station conditions. These connections between the observations and starvation-induced development remain purely hypothetical and not supported by the reported data.

(Note that the Ref. 15 the Reviewer referred to has become Ref. 14 in the revised manuscript.)

General comments:

1. Strains used: The authors selected multiple strains for different measurements. This justification is buried inside of SI and does not seem convincing. First of all, “to simplify the intercellular and cell-substrate interactions” the authors used a $\Delta pilA$ mutant, which does not make any type IV pili and thus has no S-motility (A+S-). This mutant, however, in addition to lacking S-motility this strain also produces very little extracellular matrix (ECM) which not only affects cell motility but also serves as a signaling molecule (dif signaling). This could be simply a limitation of the generality of the study, i.e. authors could just focus on aggregation in that background, but other strains that were used for particular measurements (e.g. cell tracking and polarity detection were not in the same background), e.g. $mgIB::mVenus$ strain with fluorescent label on the lagging pole was A+S+. The non-reversal mutant used was also in A+S+.

R: We have performed experiments with both the WT (A+S+) cells and $\Delta pilA$ mutant (A+S-) on agar (a stiffer substrate) and PAA (a softer substrate for traction force microscopy). In this manuscript, we focus only on the initial stage of the development, i.e. formation of a double or triple layer of cells from a monolayer. Within this regime, we did not find any qualitative difference in the behaviors of these two strains. At least, compared to the difference between $\Delta pilA$ and $\Delta frzE$, the difference between $\Delta pilA$ and WT is negligible. In the original manuscript, we compared the average properties of the WT and $\Delta pilA$ mutant in Fig. S11.

To put this in a broader context, based on our observations, the behaviors of WT (A+S+) and $\Delta pilA$ (A+S-) did not show any significant difference as single cells, thin colonies, and at the initial stage of development (formation of multiple cell layers). Only later in the development process, the difference became apparent, when the $\Delta pilA$ mutant forms long-lasting cell flows in the fruiting body that take much longer to settle down. However, $\Delta pilA$ strains can eventually form well-developed mature fruiting bodies. Our observations are supported by the literature, for example, Curtis *et al.* Journal of Bacteriology, 189(24), 9126, 2007 and Starruß *et al.* Interface Focus, 2, 774, 2012. These works reported no difference in the cluster size and no difference at the early stage when “cell towers” form (cell clusters with three layers). The Curtis paper reports that, at the early stages, cell clusters of both strains form tiers. These tiers disappear quickly in WT as cells form larger 3D aggregates, but they last longer in $\Delta pilA$ cells, which we have also observed in our own experiments. They also reported that the fruiting bodies of $\Delta pilA$ are not as round as those of WT. However, these stages of development are beyond the scope of our paper, which focuses on the mechanics of layer formation.

We agree with the reviewer that the lack of type IV pilus will affect the dif signaling. However, we have not seen any evidence showing that the dif pathway has a direct impact on the layer formation (early stages of fruiting body formation), other than tuning the reversal frequency and moving speed of the bacteria.

In the TFM experiments, we chose $\Delta pilA$ (A+S-) instead of WT (A+S+) to show in the main text because the measured traction force is easier to interpret. As discussed in the paper, to understand the mechanical forces in the system, we need to consider both the cell-substrate and cell-cell interactions. With $\Delta pilA$ cells, first, all the interactions are short-ranged: the active force can only be generated by gliding motors via direct contact; the friction and steric repulsion are contact-only as well. Second, the active forces, either between cells or between cell and substrate, can only be along the long axis of the cells. In the direction perpendicular to the bacterial bodies, there can only be passive forces, such as steric repulsion that produces pressure in the colony. These conditions facilitate our interpretation and modelling.

Figure iii. Relationship between the components of traction force \mathbf{T} and cell velocity \mathbf{v} perpendicular to the local director $\hat{\mathbf{n}}$. We fitted the middle sections of the mean data (red circles) in (a) and (b) and obtained the slopes shown in (c).

However, for the WT cells, neither condition is valid anymore. One bacterium can pull on another bacterium or the substrate at a distance using the Type IV pili. Furthermore, these interactions can be in any direction – along or perpendicular to the cell axis. The pili-mediated interactions would add two extra terms f_c^p and f_s^p to our force balance equation (Eq. 4 in the revised manuscript)

$$f_c^p + f_s^p + f_c^a + f_s^a + f_c^f + f_s^f - \nabla P = 0,$$

where superscript “p” represents pilus, “a” represents “active”, “f” represents “friction”, and subscripts “c” and “s” represent cell-cell and cell-substrate interactions, respectively. Unlike the other force terms, the forms of f_c^p and f_s^p are non-local and not subject to the body axis, which makes modelling more challenging.

To demonstrate the effects of pili, we measured the relationship between the components of traction and velocity perpendicular to the local director field, as shown in Figure iii. Relationship between the components of traction force \mathbf{T} and cell velocity \mathbf{v} perpendicular to the local director $\hat{\mathbf{n}}$. For cells without pili, the active cell-substrate traction f_s^a is longitudinal, i.e., along the director of cell

body axis \hat{n} . Hence, from Eq. 3 in the revised manuscript, the component of traction perpendicular to the director is given by

$$T_{\perp} = |\mathbf{T} \times \hat{n}| = -f_{S,\perp}^f,$$

with f_S^f being the cell-substrate friction. Therefore, T_{\perp} should be proportional to the velocity of the cells in the perpendicular direction, and the pre-factor reflects the friction coefficient. In Figure iii. Relationship between the components of traction force \mathbf{T} and cell velocity \mathbf{v} perpendicular to the local director \hat{n} . a, with $\Delta pilA$ (A+S-), although the instantaneous data is noisy, we can see a clear trend of positive correlation between the traction and velocity perpendicular to the local director. However, in Figure iii. Relationship between the components of traction force \mathbf{T} and cell velocity \mathbf{v} perpendicular to the local director \hat{n} . b for WT (A+S+), the correlation is weaker. We averaged the data (red circles) and measured the slope of the middle section. The distributions for $\Delta pilA$ (blue) and WT (orange) are shown in Figure iii. Relationship between the components of traction force \mathbf{T} and cell velocity \mathbf{v} perpendicular to the local director \hat{n} . c. For WT, some traction vs. velocity data show no positive correlation at all (slope close to zero). These results show that the forces generated by pilus do have a non-negligible effect on our force measurements, and our current force measuring technique is not capable of separating the forces generated by gliding (A) and twitching (S) motilities.

Based on these results, in the future we will investigate the effects of type IV pilus on the collective motion and mechanical forces of thin *M. xanthus* colonies. However, at this moment, the mechanics of pili-mediated interactions in this system remain poorly understood, so we avoided this complication by using the $\Delta pilA$ (A+S-) mutant.

2. Manuscript structure: As written the paper combines the results and discussion and therefore it is hard to separate authors' conclusions based on their data, speculations/interpretations, and something that is supported by the literature. Despite the large number of cited papers, it is not clear whether the cited sources support the claim; in some cases, very general reviews or several are cited for very specific claims. The authors are advised to trim their reference list and use primary literature. The SI is very loosely connected to the main text – better integration will help to strengthen the paper.

R: To separate our data from interpretations and speculations, we added a Discussion section at the end of the paper, which summarizes what we have learned and what we speculate. This reorganization of the content will help the message of the paper be conveyed more clearly. In the results sections, we now focus only on our experimental measurements, models, and derivations that can be backed up by our results.

As the reviewer suggested, we went through all the citations and trimmed off some references that are not tightly related to the claims we made. We also made sure that in every section between Introduction and Discussion, we only cited papers to support technical facts needed in our paper. We do want to point out that some citations that now appear in the Reference list are only cited in the SI. The total number of cited papers in the main text is 42, which we think is a reasonable number.

Regarding the connections between the SI and the main text, we took the reviewer's advice and strengthened their connections. For every SI figure in the Extended Data (Fig. S1 to Fig. S7), we now have somewhere in the main text pointing to them. A significant portion of our SI is used for

calibration or to demonstrate the validity of the imaging and data processing techniques. Since many of such measurements are done for the first time, such as the polarity measurement with the fluorescently-labelled MglB proteins, cell-layer thickness measurement using particle brightness, traction force microscopy under large bacterial colonies with single cell resolution, and so on, we want to convince the reader that every method has been carefully tested.

3. The first major conclusion that the authors reach via quantification of polarity and averaged local velocity is that cell flow is mainly driven by instantaneous polarity. This conclusion is achieved from the observed correlation between “local cell velocity” and “local polarity”. However, I see this correlation to be a straightforward consequence of the averaging procedure employed to compute “local cell velocity”. Authors use this argument themselves with the cartoon insets in Fig. 1f. I think due to the averaging procedure the implied term “local cell velocity” is misleading and the “cell flow” chosen in the title or cell net flux is more appropriate. I think a somewhat less trivial finding is that the presence of defect slightly changes the relationship between “local cell velocity” and “local polarity”. However, this is again can be deduced from the observations that orientation defects do not significantly slow down individual cells. If the authors agree with this simple averaging interpretation they should probably say that explicitly in this section. If the effect is stronger or weaker than simple averaging interpretation suggests that should be explicitly stated.

R: We thank the reviewer for raising this point. We agree with the reviewer that statistically, seeing high local velocity when we see high local polarity is what we expected, and this is what we illustrate in Fig. 1f. The other interesting feature of Fig. 1f is that the defects have a non-zero velocity even at zero local polarity, which is again expected based on the active nematic picture. However, Fig. 1f shows that the local velocity depends nonlinearly on local polarity. This is the main novel message of Fig. 1f. Following is a more detailed explanation.

To properly address the question raised by the reviewer, we need to compare two different models: the active nematic model and the self-propelled rod (SPR) model. In an active nematic system like a microtubule-kinesin suspension (e.g. T. Sanchez *et al.* *Spontaneous motion in hierarchically assembled active matter*. Nature, 491(7424):431–4, 2012), the flow is driven by the active nematic force, which is proportional to $\nabla \cdot \mathbf{Q}$. Therefore, flows are generated only when there is distortion in the director field (nematic order). Within such a system, each microtubule has an intrinsic polarity, but the active forces are always generated in pairs. Consequently, the flow is independent of the local polar order. In contrast, in SPR systems, the active force generated by each particle is a force vector along its polarity, so the active force and flow are closely related to the polarity field.

One interesting feature of our system is that both nematic and polar active forces coexist. Previously, we found that the average flow fields around defects can be described by an active nematic model (current Ref. 14: K. Copenhagen *et al.* *Nature Physics*, 17(2):211–215, 2021). However, by taking a closer look at the instantaneous velocity and traction force, and directly measuring the polarity in the system, we realized that its detailed behavior is much richer than what the nematic model describes. Fig. 1f helps to illustrate this point. On the one hand, if the flow in the system were driven only by active nematic forces, then we expect two flat lines shown in Figure iv, because the flow is independent of polarity. The aligned region should have $v_n=0$, because of there is no distortion in the director field. The +1/2 defect should have non-zero v_n , because there is an active nematic force due to distortion in the director field. On the other hand, if flow were driven only by the polar order, then we expect that v_n increases with p_n , and $v_n = 0$

Figure iv. Two different expected relationships between v_n and p_n . If the flow is driven by nematic active force only, then the motility of the cells in both the ordered regions and near $+1/2$ defects is independent of the polarity. Since $+1/2$ defects have a nematic active force, it generates a non-zero flow speed. If the flow is driven by polar active force only, then the velocity should be proportional to the polar order. Both lines go through the origin $(0, 0)$. Our experimental results on the right show that in *M. xanthus* cell layers, both nematic and polar active forces exist, and the latter is stronger.

when $p_n = 0$, as shown in the middle panel in Fig. iv. Considering that the cells near $+1/2$ defects experience higher friction as some of them move sideways, we expect the line for $+1/2$ defects to have a smaller slope than the aligned regions, as shown in the middle panel in Fig. iv.

What we found experimentally is neither of the two limits discussed above. It turned out to be a mixture of these two scenarios. In general, the v_n - p_n curves for both ordered regions and $+1/2$ defect show a clear positive trend. The ordered region does have a higher slope than the $+1/2$ defect. However, neither curve is straight, which means the motility is facilitated in the high polarity regime. At $p_n = 0$, v_n does not vanish near the $+1/2$ defect, consistent with the existence of an active nematic force in the system. The strong increase of local velocity with local polarity reveals that flows induced by polarity fluctuations are stronger than those induced by the active nematic force.

We agree with the reviewer that our measurement of the local cell velocity can be further improved. Velocity measurements at the level of individual cells would provide more details. However, based on our explanation above, we believe that our current measurement is sufficient to show the relative contributions of polarity fluctuations and average nematic order to the cell flows. We kept the terms “local cell velocity” and “local polarity” as what we measure is the cell velocity and polarity in small square regions ($12 \mu\text{m}$ in side length). “Local” here helps to distinguish from the global velocity and polarity of the entire colony.

4. In the same way, I find the results presented in Fig. 2 a straightforward consequence of averaging and conservation laws but they nevertheless serve as an important control in data processing.

R: We thank the reviewer for this comment. Like our answer to the Reviewer's previous question, we believe that Fig. 2 shows that the polarity field is the main driver of the flows leading to layer formation. The physical pictures are different between flows and fluxes driven by either nematic order or polar order, although they can both lead to layer formation. As the reviewer points out, we made use of mass conservation in Eq. 2 to calculate the volume change based on the velocity flux. However, what really matters is our direct measurement of the polarity flux, which is not derived from a conservation law. Our results show that instantaneous polar activity is the main force that drives influx of cells, not the nematic order, which is not a trivial result. Polarity-induced fluxes could have been small compared those produced by the active nematic force, but we found they instead dominate.

To illustrate this difference, we added a schematic diagram in Fig. 2, which is the current Fig. 2a. This is also related to our response to the Comment 4 raised by Reviewer #1.

The reviewer's comment made us realize that we did not explain the significance of this measurement well in our previous manuscript. We have added some more explanation to clarify the physical picture and our motivation in the section *Polarity-driven cell influx drives layer formation*.

5. I found the results of traction force microscopy measurements the most novel and interesting aspects of the work. Interestingly, the nematic theory can predict the average traction forces despite the huge std deviations. Can authors comment on why this is the case? Can the theory say something about the fluctuations? What is the major source of the fluctuations – defect-to-defect differences? Or stochasticity of cell behaviors? It may be worth discussing in the main text how the theory used here differs from that presented in Ref. 15.

R: We agree with the reviewer that it is nice to see that the averages are well captured by our active nematic model despite the huge fluctuations. Given that the average order is nematic, the long-time average of the forces and flows should be captured by an active nematic model. Due to the cell reversals, the instantaneous polarity of the cells cancels each other in the long term and yields no average polarity. The Reviewer also asked in the summary at the beginning, "why the mean behavior still agrees with the nematic theory developed in Ref. 15 (Ref. 14 in the revised manuscript)". The answer is the same.

We believe that the main source of fluctuations is stochastic cell reversals. The traction generated at each defect at any time is controlled by its nematic order and the instantaneous polarity of the bacteria near the defect at that moment. We have added a detailed discussion on the origins of fluctuations in the system and how we understand them. Different defects do not show significant differences. However, within each defect, the traction and velocity fluctuations are always large. These fluctuations happen on short time scales (seconds) and appear instantaneous with the temporal resolution of our measurements — the next frame can be very different from the previous frame. Beyond the cell polarity, further stochasticity in cell behaviors can bring extra fluctuations, which provides an "intrinsic noise" in living systems that does not exist in synthetic active materials, caused by cells sensing and responding to the local environment and regulating their individual behaviors. The consequences include a distribution of cell activity, reversal frequency, and so on. This is a very interesting topic to explore further, but it is beyond the scope of this paper. We comment on this point in the new Discussion section.

Our current model is still a nematic model that builds on Ref. 14 [K. Copenhagen *et al.* Nature Physics, 17(2):211–215, 2021]. As a result, it does not predict anything about the fluctuations, which remains a challenge for future work. The main difference between our model and that of Ref. 14 is that here, we imposed a constraint on the velocity divergence $\text{div } v$ to match the experimental measurements.

6. Again, from the averaging perspective higher polar order in cells that do not reverse is not too surprising. However, I found the author's attempt to connect the results to the increased polar order of the system with starvation-induced aggregation unconvincing. The importance of reversals on the collective cell alignment and motility patterns has been demonstrated in the earlier publications of Shaetitz lab (<https://doi.org/10.1098/rsif.2015.0049>) and by others (<http://dx.doi.org/10.1098/rsfs.2012.0034>). These results demonstrate that nearly complete elimination of the reversals in ΔfrzE would be quite distinct from a relatively small (on average) and highly heterogeneous increase in reversal times at the onset of the developmental aggregation. Moreover, under non-starvation conditions, non-reversal strains can persist for a long time without forming fruiting-body-like aggregates indicating that the formation of the multiple layers observed in the defects and formation of fruiting bodies may not be controlled by the same mechanism. Perhaps a better way to assess how moderate perturbations in average reversal frequency affect the polar order and layer formation would be to use chemical attractants or repellants that are known to modulate the reversal period of wild-type cells (casitone, isoamyl alcohol etc. see e.g. <https://doi.org/10.1046/j.1365-2958.1997.4261783.x>).

R: While there are differences between changes in reversals seen during development and the FrzE mutant which nearly lacks reversals, we disagree that this is not a relevant model. As we reported in Liu et al. PRL 122(24) 248102 (2019) and Guannan Liu's PhD thesis, Myxo cells alter their speed and reversals from $v \sim 0.5 \mu\text{m}/\text{min}$ and a $T_{\text{rev}} = 6 \text{ min}$ before starvation to $v \sim 2.5 \mu\text{m}/\text{min}$ and $T_{\text{rev}} = 12 \text{ min}$ at the point when significant aggregation is observed. In an active matter context, the Peclet number is the most relevant quantity to describe the dynamics. Alternatively, it is also useful to think of the reversal period in terms of how many body lengths a cell travels before it reverses. For the pre-starvation values, the cell travels 3 microns on average before reversing, which is significantly less than the length of a cell. On the other hand, cells travel ~ 30 microns before reversing after starvation, which is ~ 6 body lengths. Thus, relative to a body length, the starving WT cells and FrzE mutant cells are thus both in the "long" reversal period regime by a large margin, and hence they can be compared.

Starruß et al., referenced by the reviewer, found similar results, reporting that "During fruiting body formation the reversal frequency decreases up to a point where cell movements become nearly unidirectional" and "Our results suggest that only by switching on and off the reversal can cells modify dramatically their collective behaviour, with the suppression of cell reversal leading to collective motion in the form of moving clusters and vortex formation at high densities. This observation is consistent with the observed decrease in reversal frequency in the wild-type upon nutrient depletion, which is followed by the formation of large moving clusters and aggregation of cells." We therefore think that our comparison of the differences between the WT cells and the FrzE mutant in nutrient rich conditions and the events that take place in the early stages of starvation-induced aggregation is fair.

The reviewer also asks whether the increase in layer formation that we see in the FrzE mutant in nutrient-rich conditions can be compared to starvation-induced fruiting body formation in the WT

Figure v. Three-dimensional surface topography image sequence of the formation of a single fruiting body in a standard *M. xanthus* starvation assay. Color indicates height in the third dimension and images are shown with a period of one hour. The colony starts as a single 2D layer and, upon starvation, cells form three-dimensional aggregates through layering over the course of hours. At least 8 layers can be identified in the data, and even the fully mature fruiting body (5hr) contains clear layering near its base.

given that this mutant does not form aggregates in our experiments. We would like to emphasize that we only focus on the very initial stage of fruiting body formation. We argue that fruiting bodies are seeded by layer-formation events triggered by the mechanical forces measured in our work. At a later stage, a fraction of the multi-layered cell aggregates will keep on developing into droplet-like fruiting bodies (see Figure v). At these later stages, cell behaviors and cell-environment interactions may change significantly, but these effects are beyond the scope of this paper.

Finally, we have found the chemicals mentioned by the reviewer to be unreliable at controlling reversal frequency quantitatively. We have not been able to accurately control reversals in the absence of other factors (such as nutrient level) with these compounds and thus we chose not to use them in our experiments.

7. In light of the above and given that none of the experiments were performed under starvation-induced development, I would strongly suggest removing “developing” from the title and other places in the manuscript. Conclusion “we show that polarity fluctuations trigger the formation of new cell layers, which enables the starvation-induced development from monolayers to droplet-like fruiting bodies in the social bacterium *Myxococcus xanthus*” advertised in the intro is similarly unjustified.

R: We thank the reviewer for raising this point. As discussed in the previous comment, the purpose of this research is to show that to trigger cell aggregation via layer formation, one could use the reversal frequency as the sole parameter without the necessity to consider any other more complex mechanisms and biological regulations. The logic behind this work is this: As the *M. xanthus* colonies develop, they form three-dimensional cell aggregates. These large cell aggregates originate from cells creating layered structures, and to achieve that, the cells need to be able to move out of plane to form double or triple layers from a monolayer. Indeed, aggregation proceeds in our hands up to eight or more layers, beyond which the curvature of the fruiting body makes it hard for us to observe layering (Figure v). In the literature, people have reported that development starts from “cell towers”, which are double or triple cell layers, and the three-dimensional structure builds up layer by layer, until at some point it rounds up (Curtis *et al.* Spatial Organization of *Myxococcus xanthus* during Fruiting Body Formation. 189(24), 9126, 2007). At the same time, the developmental process starts as the cells lower their reversal frequency. Thus, our hypothesis is that by changing their reversal frequency, the cells change the active-force fluctuations within the cell layer, which triggers the spontaneous formation of new layers. We think this justifies our arguments and conclusions involving “development”.

We did perform traction force microscopy with cells in a nutrient-poor environment, and we confirmed that *M. xanthus* can form fruiting bodies on the surface of a very soft substratum such as PAA. However, we found that the nutrient-poor liquid medium TPM has a significantly higher surface tension compared to the nutrient-rich medium CTTYE. As a result, the cell movement in TPM shows some visible differences from that in CTTYE. Using different nutrient conditions may also cause other changes in cell behaviors, metabolism, etc. As a result, we believe that using $\Delta frzE$ cells in a nutrient-rich environment is actually a better-controlled experiment than performing the measurements using WT in a nutrient-poor environment.

As suggested by the reviewer, we have removed the word “developing” from the title to avoid causing the readers to think that we are talking about a later developmental stage rather than the early stage with multi-layer formation. We also emphasize that we are focusing on the early stage of development – double layer formation. We have revised the introduction, including the sentence that the Reviewer pointed out, and the section *Cell reversals control local polar order and layer formation*. Now we clearly state what our aim is and why we chose $\Delta frzE$ as the model system, supported by the literature.

Reply to the reviewers' comments on **Local polar order controls mechanical stress and triggers layer formation in *Myxococcus xanthus* colonies**

Endao Han, Chenyi Fei, Ricard Alert, Katherine Copenhagen, Matthias D. Koch, Ned S. Wingreen, and Joshua W. Shaevitz

We thank all the reviewers for their positive feedback. Reviewer #1 and Reviewer #2 made no further comments. Here, we address the comments and questions raised by Reviewer #3. The original comments are in **blue**, and our responses are in **black**. The revisions in the original manuscript are highlighted in **blue**. The figures in this reply are labelled as Fig. i, Fig. ii, and so on to avoid confusion with the figures in the manuscript, which are labelled as Fig. 1, Fig. 2, etc.

Reviewer #1 (Remarks to the Author):

I find the authors have addressed the most of the comments raised by the referee's and also made the required modification in the manuscript. Hence now I feel manuscript is suitable for publication in Nat. Comm.

Reviewer #2 (Remarks to the Author):

I thank the authors for their response. Their clarifications and revisions have addressed my comments and I recommend publication of the manuscript.

We thank the Reviewers #1 and #2 for their positive feedback.

Reviewer #3 (Remarks to the Author):

After reading the author's response letter and revisions, I still have three points concerning the results and interpretation.

1) Regarding the fundamental nature of the active nematic and the self-propelled rod models in application to the description of the myxobacteria: In my interpretation, the active nematic model is fundamentally phenomenological (hydrodynamic description). It can be formulated by the symmetry arguments based on the postulated energy functional or derived from the microscopic description under certain limits and assumptions (<https://arxiv.org/html/2401.05070v1>). Indeed, in the previous manuscript by some of the authors (Ref. 14), it was shown that their data on the average flow fields around defects is consistent with a particular form of the active nematic model. However, no microscopic justification for this has been provided. The authors answer my question on this with a somewhat superficial 'averaging' argument. However, this averaging is not trivial, especially in a system with strong and non-linear interactions, and when the time/length scales of the mesoscopic phenomena (defect) and the microscopic phenomena (free path of a cell) are of the same order. On the microscopic scale, unlike the microtubule system, the authors reference in their response or the preprint above, myxobacteria cells are polar. Therefore, the self-propelled rod model is a more appropriate mechanistic behavior. Thus, I still think that interpreting the importance of the polar order based on the breakdown of the averaging assumptions required is the most obvious interpretation of the results and is totally expected. From this perspective, the averaging is even more likely to fail if agents stop switching their polarity. From the same perspective, the equations and theories presented in the main text need to be explained/defined

with more rigor. For example, it is not clear if the force balance in Eq. (4) represents a force balance on the microscopic scale (single cell) or mesoscopic (hydrodynamic small volume with many cells).

To summarize, given the fundamentally polar nature of force generation in moving cells, it is by no means surprising that polar effects are significant and lead to large fluctuations from the active nematic description. The authors claim these fluctuations are “anomalously” large, but their “anomalous” nature has not been demonstrated. It is even less surprising that polar effects are more important when cells do not reverse. What is surprising is that at topological defects, the average cell velocity (result in Ref. 14) and active force (new here) agree with predictions from active nematic theory. But that effect remains unexplained.

R: We agree with what the Reviewer wrote in the first half of the comment. The fact that the average velocity and traction in the regions around the half-integer defects can still be well described by the active nematic theories, even with strong polarity fluctuations, does not have a trivial explanation. However, that is not the focus of this paper. The goal of this paper is to point out the limitations of the active nematic theories in describing systems like this, and highlight the phenomena that they do not predict, such as the large traction and velocity fluctuations near the defects, the formation of cell double layers with a much faster instantaneous rate than predicted, and the effects of the cells’ reversal frequencies on the colony morphology. We also agree with the Reviewer that systematically coarse-graining a microscopic model of self-propelled rods would be a very interesting approach to capture the large-scale behavior of our experimental system. But again, our major aim here is to provide quantitative evidence showing why it is necessary to push beyond the current theories of active nematics. We do not intend to present such a new theoretical framework in this paper.

We also agree with the reviewer that the findings agree with what you would expect for a system of self-propelled rods. However, we do not think that when a system has both nematic order and cell polarity, it is obvious to predict if it will behave more like active nematics or self-propelled rods, as will be discussed in detail below. We also believe that providing experimental evidence of strong polarity fluctuations in a system with nematic order is important progress.

- There is a lack of experimental studies reporting polarity fluctuations in active nematics. We provide experimental evidence for such fluctuations, and we show that they have an important biological role: they trigger the formation of cell layers.
- Second, previous work emphasized the importance of the average nematic order and the resulting flows near half-integer defects, and such measurements were mostly about the averaged properties. Thus, cell aggregation is expected to be driven by these average flows and to happen at an approximately constant rate. Indeed, that is what happens in many systems. Fig. i is adapted from Kyogo Kawaguchi et al. Nature 545, 327–331 (2017) and its Supplemental Movie. The authors performed experiments with neural progenitor cells on a solid substrate and showed that the cells aggregate at $+1/2$ defects with a relatively constant rate (they showed only two experiments in the paper). In this system, the instantaneous behavior is similar to the “average” behavior predicted by the nematic theory, with some fluctuations. In contrast, our paper shows that the fluctuations in *M. xanthus* play a vital role as they trigger layer formation events. For *M. xanthus* colonies, the previous approach can only explain the statistical behaviors when averaged over a long time, but it does not capture fast processes like layer formation events, which we now show are driven by polarity fluctuations.
- Third, based on our review of the literature, our results are different from most published “expectations”. In papers studying self-propelled rods, they rarely discuss the topological defects, while in the systems with half-integer defects, normally an active nematic theory

works well as many people have reported. One exception is the paper by Takuro Shimaya and Kazumasa Takeuchi, PNAS Nexus, 1, 1-11, (2022). In a growth driven bacterial colony, they found that the active nematic theory makes a prediction that lightly deviates from their experimental measurements. So, they introduced cell polarity, due to the asymmetric friction applied on the bacteria by the substrate when the bacterial cells are slightly tilted (one end on the substrate but the other moves up). They found that the corrected model makes a better prediction, as shown in Fig. ii. However, the effects of polarity in their system are very limited. The cell flow is still driven by the nematic active force. Again, this is the physical picture most people expect when studying systems with half-integer defects.

- Fourth, in *M. xanthus*, the strength of polar fluctuations compared to the average nematic forces depends on the cell reversal frequency. Would the polar fluctuations be relevant for the reversal frequencies used by *M. xanthus*? We have shown that for reversal periods of several minutes, the effects of individual cell polarity are already very strong in our system. Furthermore, *M. xanthus* cells can glide on the solid substrate and glide on each other. The former leads to a polar active force because its equal and opposite force is applied on the substrate, which is not a part of the active system. In contrast, the latter interaction leads to a pair of equal and opposite forces both applied on the cells within the colony, which introduces a nematic active stress. It was unclear which component plays a more important role in the dynamics of our system, and our work shows that the fluctuations associated with the polar component are strong and key for layer formation.

In summary, we believe that one of the significant breakthroughs made by this work is that, for the first time, it quantifies the relative contributions of the nematic order and polarity in a motility-driven active system with both features. We showed the effects of these fluctuations on several observables: velocity and traction fluctuations near the defects, rate of layer formation, and the traction power spectrum. While it is expected that cell polarity will do something, it is not trivial to predict how strong the effects are, and our measurements provide evidence for the magnitude and biological relevance of polar fluctuations in systems of self-propelled rods.

Regarding the Reviewer's question about the model, in this paper, we model the system as a continuum, so in Eq. (4), the forces are in the forms of a field. Eq. (4) therefore provides a mesoscopic force balance.

Lastly, we do agree with the Reviewer that the word "anomalous" in our abstract is misleading, because there is no estimation of how large "normal" fluctuations are. We thank the Reviewer for raising this point. In almost all the other experimental papers, the standard deviations of the velocity and the orientational order near defects are not shown, so we cannot compare our measurements with others. We previously used the word "anomalous" to mean that the fluctuations are larger than the mean. To fix this issue, we have edited the abstract, so it now reads: "Average cell velocity and active force at topological defects agree with predictions from active nematic theory, but their fluctuations are substantially larger than the mean due to polar active forces generated by the self-propelled rod-shaped cells."

2) I find data on Fig. iii in the response very interesting and recommend its inclusion into the manuscript. It does not fully alleviate my concern about changing from strain to strain in different measurements but perhaps supports it further. I cannot fully follow the theoretical arguments presented. Suppose the force balance equation (4) is interpreted in a hydrodynamic sense. In that case, the pili forces between cells in that averaging volume should vanish or be absorbed into a negative pressure term, and the forces from pili extending away from the volume should, on average, be in the direction of the nematic alignment. And given that traction forces for WT cells do not completely agree with the active nematic model (is it?), it is surprising that in Ref. 14 for WT, average cell velocity nevertheless matches that theory.

R: As suggested by the Reviewer, we have added the Fig. iii in the previous reply to the Supplemental Information of the revised manuscript. This is what it looks like in the manuscript:

Fig. S12: Effects of type-IV pili on the instantaneous traction force generated by the bacteria. (a,b) Relationship between the components of traction force \mathbf{T} and cell velocity \mathbf{v} perpendicular to the local director $\hat{\mathbf{n}}$. These perpendicular components $|\mathbf{T} \times \hat{\mathbf{n}}|$ and $|\mathbf{v} \times \hat{\mathbf{n}}|$ are measured point by point in the videos. The color map shows the normalized number density of a location with certain $|\mathbf{T} \times \hat{\mathbf{n}}|$ and $|\mathbf{v} \times \hat{\mathbf{n}}|$. (a) is obtained with the $\Delta pilA$ strain and (b) is with the WT. We fitted the middle sections of the mean data (red circles) in (a) and (b), and did this for all the videos that we took, and obtained the slopes shown in (c). The ratio between $|\mathbf{T} \times \hat{\mathbf{n}}|$ and $|\mathbf{v} \times \hat{\mathbf{n}}|$ provides a viscous-like friction coefficient. The results obtained with the $\Delta pilA$ cells clearly deviate from zero, which means that the transverse components of velocity and traction are correlated. This is consistent with transverse tractions being due to just viscous friction, with no contribution from active forces. In the absence of pili, active forces arise from cell gliding, and they are therefore along the cell body axis, and hence along the director $\hat{\mathbf{n}}$. However, for the WT cells, the correlation between transverse traction and velocity becomes significantly weaker because now a cell can potentially pull on its lateral neighbors using type-IV pili and thus generate active forces perpendicular to its long axis. This effect covers up the relationship between $|\mathbf{T} \times \hat{\mathbf{n}}|$ and $|\mathbf{v} \times \hat{\mathbf{n}}|$ given by just cell-substrate friction. To avoid this complication, we used the $\Delta pilA$ strain instead of the WT as the model organism in this paper. Nevertheless, we measured velocity and traction fields around defects in WT colonies, as shown in Fig. S14.

Fig. S14: Director, velocity, traction, and traction fluctuation fields around $\pm 1/2$ defects in thin colonies of WT cells (A^+S^+ strain with reversals). (a) Experimentally measured mean director field $\langle \hat{n} \rangle$ around $\pm 1/2$ defects. (b) Experimentally measured mean velocity field $\langle \mathbf{v} \rangle$ near $\pm 1/2$ defects. The black arrows show its magnitude and direction and the color map shows the speed $|\langle \mathbf{v} \rangle|$. (c) Experimentally measured mean traction field $\langle T \rangle$ around $\pm 1/2$ defects. The color maps show their magnitudes, and the arrows indicate their magnitude and direction. (d) Experimentally measured standard deviation of traction, σ_T , around $\pm 1/2$ defects. The black lines show the director field $\langle \hat{n} \rangle$.

We agree with the Reviewer that incorporating pili-generated forces in our model is an open problem, which is why we chose the ΔpilA mutant without pili. The pili normally extrude out of the leading pole of the cell and can extend longer than a cell length. They can attach to the substrate and to other cells, and they generate a pulling force. So, it could act as a contractile active force that pulls the cells closer to each other in both the tangential and transverse directions. However, we know neither the number of active pili per cell nor their pulling force or pulling rate in the conditions of our experiments. In our study, we therefore decided to focus on mutants without pili, and we defer the study of pili-generated forces to future work.

Yet, to study the role of pili, we performed traction force microscopy experiments with WT cells and analyzed the average velocity, average traction, and standard deviation of traction near the $\pm 1/2$ defects, as shown in Fig. S14 above. Here, the data are obtained with about 600 defects so the statistics is not as good as those for ΔpilA in the main text. However, both data sets show similar patterns and magnitudes. Especially, the traction fluctuation (standard deviation) is comparable to the ΔpilA mutant and far less than the ΔfrzE mutant that does not reverse. We have added this figure to the SI of the manuscript as well. It shows that for all the phenomena that we discuss in this paper, the ΔpilA (without pili) and WT (with pili) strains show similar results.

We have also edited the text in the Section III A of the SI accordingly.

3) The authors continue (but to a lesser extent) to imply that their results are relevant for the multi-layer formation in starvation conditions. For example, in the intro, the paragraph starting with “In this work” continues with the sentence, “As nutrients become scarce, the cells increase τ and start to form double or triple layers on top of the original monolayer.” This has not been demonstrated here, and the causal effects of these phenomena remain questionable. It should be noted that (1) the changes in reversal frequency during starvation often happen in the later stages of development after the initial “cell towers” have been established; (2) Development often experimentally observed at cell densities that implied initially multi-layer situation; (3) The extent of changing the reversal frequency cited is not consistent with other pubs (<https://doi.org/10.1128/jb.00544-07>, <https://doi.org/10.1073/pnas.1620981114>)

R: First, the sentence in the introduction pointed out by the Reviewer is not based on our own results reported in this paper, but on existing results in the literature (Curtis et al. *Journal of Bacteriology*, 2007 and Liu et al. *PRL*, 2019). We have added the references in that sentence to avoid confusion. Furthermore, in Fig. 4a,b, we showed that a mutant with lower reversal frequency forms more layers. So, even though we do not provide direct measurements with the cells under starvation (because of the reasons stated in the previous response to reviewers), we believe that our results in Fig. 4 suggest that the decrease in reversal frequency during starvation could promote layer formation. We are by no means excluding any other mechanisms that contribute to the complex process of fruiting body formation, including in the early stages. We are simply suggesting a physical mechanism that could also contribute to this process, thus helping in the initial stages of fruiting-body formation.

Regarding the three points raised by the Reviewer:

- (1) First, as discussed earlier, the formation of cell towers is not a steady accumulation process where these towers are initiated and keeps growing. The system has strong fluctuations before these towers appear. Second, according to the literature (Curtis et al. *Spatial Organization of Myxococcus xanthus* during Fruiting Body Formation. 189(24), 9126 (2007)), for WT cells, no apparent cell towers are present at 3 hours post-starvation, and more persistent cell towers can form 8-10 hours after starvation. According to the measurements in our lab shown in Fig. iii (Liu et al. *Self-Driven Phase Transitions Drive Myxococcus xanthus* Fruiting Body Formation. *PRL*, 122, 248102 (2019) and Guannan Liu's PhD Thesis (2019)), the reversal frequency starts to decrease 4 hours after starvation and might reach a steady value at 9 or 10 hours post-starvation. According to this data, the decrease in reversal frequency seems to occur around the same time as the formation of layered structures.
- (2) We agree with the Reviewer that the cell concentration on the surface is a very important parameter controlling colony development. In our experiments, we chose a cell concentration that consistently produces densely packed cell monolayers, but not multilayers. For a cell monolayer we can directly connect the traction to the cell orientations and flow. Otherwise, if the colony has multiple layers to begin with, the cell-cell interactions are more complicated, and measuring cell orientation and velocity is more difficult. Both in our experiments and in Ref. [30], the colony develops by forming layers and fruiting bodies despite starting as a monolayer. Therefore, development can happen at cell densities that do not imply multilayering.
- (3) We have read the two papers suggested by the reviewer carefully, but we have different interpretations of their results. The first paper by Berleman and Kirby focused on the predation behavior of *M. xanthus*, which is a very different condition from starvation. We also did not

find their measurements of the change in reversal frequency as a function of time. The second paper by Cotter et al. seems to agree with our results. The right panel in Fig. iii is from this paper. After 11-12 hour of starvation, there is a strong correlation between increased duration between reversals (inverse of reversal frequency) of the cells in the colony and the initial aggregation of cells. The time is also consistent with the measurements carried out in our lab, which is also shown in Fig. iii. Overall, we do not see anything in these two papers that is inconsistent with our results or that opposes our conclusions in this paper.

In conclusion, although we appreciate the Reviewer's suggestions, we believe that our results about the non-reversing mutant are informative about the early stage of development and fruiting body formation.

To be clear – these concerns are mainly not about the data presented but about its interpretation, so I don't want my feedback to stand in the way of paper publication.

R: We thank the Reviewer for the constructive suggestions and comments. We hope our revisions are satisfactory and our answers have addressed all the concerns.